# Layer-optimized SAR processing with a mobile phase-sensitive radar: a proof of concept for detecting the deep englacial stratigraphy of Colle Gnifetti, Switzerland/Italy

Falk M. Oraschewski[1], Inka Koch[1], M. Reza Ershadi[1], Jonathan Hawkins[2], Olaf Eisen[3,4], and Reinhard Drews[1]

[1]University of Tübingen, Department of Geosciences, Tübingen, Germany
[2]Cardiff University, School of Earth and Environmental Sciences, Cardiff, Wales, United Kingdom
[3]Alfred Wegener Institute, Helmholtz Centre for Polar and Marine Research, Bremerhaven, Germany
[4]University of Bremen, Department of Geosciences, Bremen, Germany

**Correspondence:** Falk M. Oraschewski (falk.oraschewski@uni-tuebingen.de)

**Abstract.** Radio-echo sounding is a standard technique for imaging the englacial stratigraphy of glaciers and ice sheets. In most cases, internal reflection horizons (IRHs) represent former glacier surfaces and comprise information about past accumulation, ice deformation and allow to link ice core chronologies. IRHs in the lower third of the ice column are often difficult to detect or coherently trace. In the polar ice sheets, progress in IRH detection has been made by using multistatic, phase-coherent radars, enabling focused synthetic-aperture radar (SAR) processing. However, these radar systems are often not suitable for deployment on mountain glaciers. We present a proof of concept study for a lightweight, phase-coherent, and ground-based radar system, based on the phase-sensitive radio echo-sounder (pRES). To improve the detectability of IRHs we additionally adapted a layer-optimized SAR processing scheme to this setup. We showcase the system capability at Colle Gnifetti, Switzerland/Italy, where specular reflections are now apparent down to the base of the glacier. Compared to previously deployed impulse radar systems, with the mobile pRES the age of the oldest continuously traceable IRH could be increased from $78 \pm 12$ a to $288 \pm 35$ a. Corresponding reflection mechanisms for this glacier are linked to stratified acidic impurities which in the upper part were deposited at a higher rate due to increased industrial activity in the area. Possible improvements of the system are discussed. If successfully implemented, these may provide a new way to map the deep internal structure of Colle Gnifetti and other mountain glaciers more extensively in future deployments.

## 1 Introduction

Polar mid-latitude glaciers store information about past regional climate change (Wagenbach et al., 2012) and hold natural (Clifford et al., 2019) and anthropogenic (Gabrieli and Barbante, 2014; Sigl et al., 2018a) impurity records that can be extracted by ice coring. Being located in the vicinity of highly industrialized regions (Sigl et al., 2018a), these archives are an important complement to ice core records from the polar ice sheets spanning deep time (EPICA community members, 2004; NEEM community members, 2013). However, the interpretation of these records can be limited by uncertainties in ice core dating (Jenk et al., 2009). Using radar surveys to laterally trace internal reflection horizons (IRHs) between multiple ice core sites,

their chronologies can be compared to reduce uncertainties (Eisen et al., 2003; Bohleber, 2011; Licciulli et al., 2020). IRHs represent discontinuities of the dielectric permittivity or conductivity between different layers of ice. Permittivity is controlled through density and crystal-orientation fabric (COF), whereas conductivity is determined through acidity (Fujita et al., 2006).

IRHs are isochronous when they are formed by seasonal snow-density variations or impurities that were initially deposited at the surface. Besides supporting the ice core dating, IRHs can be used to infer spatial accumulation patterns (Cavitte et al., 2018; Koch et al., 2023), past ice deformation (Drews, 2015; Koutnik et al., 2016), upstream effects for ice core records (Eisen et al., 2003), and for finding new ice-core sites with an intact stratigraphy (Lilien et al., 2021). In many of these applications the deep and old stratigraphy is of particular interest.

Both on mountain glaciers (Eisen et al., 2003; Konrad et al., 2013) and on polar ice sheets (Drews et al., 2009) the lowest third of the ice column is often difficult to image, for example due to increased radio-wave attenuation, weakening of density contrasts, or buckling and folding of reflection interfaces. For polar ice sheets, the emergence of phase-coherent radars and linked synthetic-aperture radar (SAR) processing have significantly improved the detection of the deep englacial stratigraphy (Hélière et al., 2007; Peters et al., 2007). These heavy and power-intensive systems are pulled by tracked vehicles (Paden,

2006), or mounted on aircraft (Shi et al., 2010), neither of which are applicable on many mountain glaciers. Instead, smaller, typically incoherent, impulse ground penetrating radars (GPR) available in a variety of off-the-shelf products are used. Here, we address the need for a lightweight, low-power, phase-coherent and ground-based system suitable for SAR processing.

SAR processing improves the along-track resolution and suppresses clutter in the radargram by coherently focusing the backscattered power of point targets illuminated across multiple traces (Peters et al., 2005; Kusk and Dall, 2010). Similar

to migration, this will collapse along-track hyperbolas from off-nadir reflections, providing improved imaging of, e.g., basal structures. However, in the case of specular reflections from IRHs, the signal quality can also deteriorate during SAR processing due to destructive interference (Holschuh et al., 2014). Castelletti et al. (2019) addressed this shortcoming by introducing a layer-optimized SAR (LO-SAR) processing method that corrects for along-track phase shifts of specular reflections before coherent summation. Other approaches for improving the detection of IRHs are based on filtering the IRHs contribution in

the azimuth frequency domain of the radargram (Heister and Scheiber, 2018) or on including spatial correlation information into SAR processing (Xu et al., 2022). In addition, most of these algorithms determine the slope of IRHs, which can support the automated tracing of IRHs (MacGregor et al., 2015) and provides an alternative metric for radar data–model comparison (Holschuh et al., 2017).

In this study, we showcase a mobile, phase-coherent radar system that is suitable for SAR processing and can be deployed

on mountain glaciers. We use the existing phase-sensitive radio echo-sounder (pRES, also ApRES for autonomous and stationary surveying), a lightweight, low-power and inexpensive frequency modulated continuous wave (FMCW) radar operating at 200–400 MHz (Brennan et al., 2014; Nicholls et al., 2015). This instrument has been widely used, most prominently in determining basal melt rates beneath ice shelves (Vaňková et al., 2021; Zeising et al., 2022) for which it was originally designed. Ershadi et al. (2024) presented a mobile acquisition of polarimetric pRES data using an autonomous rover to measure the COF

on large spatial scales with postings of several tens of meters. The feasibility of using the pRES for profiling with sub-meter spacing has only theoretically been assessed by Kapai et al. (2022), who investigated artifacts that can arise from moving the

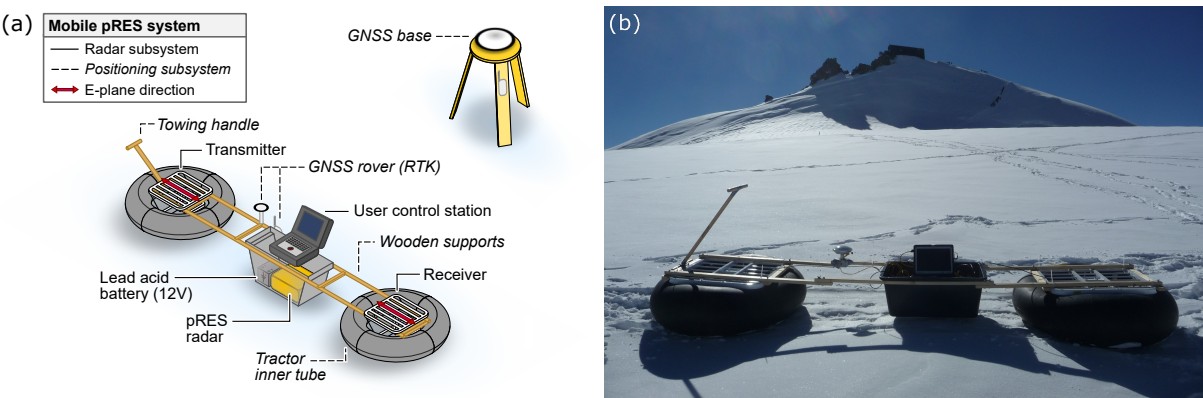

**Figure 1.** Mobile pRES setup with (a) a schematic of its components and (b) a photograph of the deployment on Colle Gnifetti.

pRES during data acquisition. Here, we present the first mobile deployment of the pRES for profiling and demonstrate its capability to detect deep IRHs that were unresolved in previous surveys. In the following we first outline the mobile pRES setup and the data acquisition approach (Section 2), before describing the applied FMCW signal and LO-SAR processing (Section 3). In a proof of concept survey, mobile pRES radargrams are compared to GPR data from impulse systems and to available ice core data to interpret the imaged signatures in the glaciological context (Section 4). Finally, limitations of the mobile pRES are discussed including suggestions for further developments (Section 5).

## 2  Mobile pRES

Coherent profiling with the pRES requires decimeter scale postings with a relative positioning accuracy on the centimeter level. Here, we describe the hardware design and data acquisition approach for mobilizing the pRES, as applied in our proof of concept study (Section 4). Potential improvements to this system are discussed in Section 5.2.

### 2.1  Hardware design

To mobilize the pRES for profiling, we placed the transmit and receive skeleton slot antennas in inflated tractor inner tubes with the antennas being separated by $2.7\,\mathrm{m}$ and elevated a few centimeters above the snow surface (Fig. 1). Both antennas were oriented such that the E-plane was parallel to the profiling direction (also denoted as HH orientation). Positioning was controlled by the Trimble R9s global navigation satellite system (GNSS) receiver, operated in real-time kinematic (RTK) mode with the baseline between the GNSS base station and rover being typically below $200\,\mathrm{m}$. This setup provided the relative position of the pRES to the base station with centimeter accuracy in real-time. Both, the positioning system and the pRES (operated in 'attended mode') were connected to a control station for automatic triggering of radar signal acquisition based on the position.

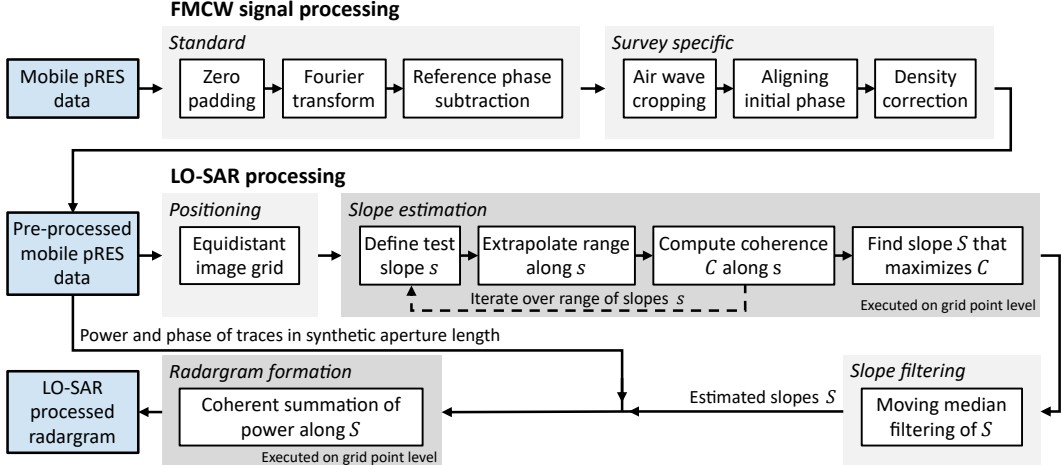

**Figure 2.** Overview of FMCW signal and LO-SAR processing scheme. Blue boxes represent the radar data at different processing stages. Gray blocks summarize the main processing steps, where a darker gray indicates that these are executed on the grid point level.

## 2.2 Data acquisition

In our survey, we used the standard configurations of the pRES, including a chirp time of $1\,\text{s}$. This makes data collection in stop-and-go mode necessary, because moving the pRES during signal recording can cause Doppler blurring (Kapai et al., 2022), promote incoherent backscatter and increase system noise from cable movement. We therefore aimed to minimize the stopping time per point by first recording two and later only one chirp per point/trace. When two chirps were recorded, only the one with the lower noise floor was selected for further processing. Using the control station, chirping was triggered once the system was hand-towed by more than $9\,\text{cm}$ and audible signals indicated the operator to start/stop towing. With this approach, an average trace spacing of approximately $\Delta d = 13\,\text{cm}$ and a median data acquisition time of $10\,\text{s}$ per trace was achieved, giving a total survey speed of around $3\,\text{h}/100\,\text{m}$.

## 3 Data processing

This section outlines the LO-SAR processing routine (Fig. 2), starting with the standard FMCW signal processing and survey-specific adaptations which provide the background for estimating IRH slopes from the phase signal during the subsequent LO-SAR processing.

### 3.1 FMCW signal processing

The description of the standard FMCW signal processing and the notation broadly follows Brennan et al. (2014). The pRES transmits a linearly frequency modulated chirp with a center frequency of $f_\text{c} = 300\,\text{MHz}$ and a bandwidth $B = 200\,\text{MHz}$ over a chirp time of $T = 1\,\text{s}$. On the hardware side, the reflected signals are mixed with the transmitted signal and subsequently

low-pass filtered. This results in the deramped waveform which, by using a fast Fourier transform (FFT), is decomposed into the deramped frequencies

$$f_{\mathrm{d}} = \frac{2B\tilde{R}\sqrt{\varepsilon_{r,\mathrm{ice}}}}{Tc},$$ (1)

which relate to the reflection range $\tilde{R}$ using the speed of light in vacuum $c$ and the real-valued relative permittivity of ice $\varepsilon_{r,\mathrm{ice}} = 3.17$ (Brennan et al., 2014). Note that this formulation does not allow the use of a variable relative permittivity and the reflection range does not consider the lower density of firn (indicated by $\sim$). A density correction will be described below and from here on, the general relative permittivity of firn or ice $\varepsilon_r$ is used.

The range resolution $\delta_{\mathrm{R}}$ of FMCW radars is determined by their bandwidth

$$\delta_{\mathrm{R}} = \frac{c}{2B\sqrt{\varepsilon_r}},$$ (2)

giving a resolution of the pRES of $\delta_{\mathrm{R}} = 0.42\,\mathrm{m}$ in ice (Brennan et al., 2014). For each frequency component, or range bin $n$, the FFT returns the amplitude $a_n$ (Fig. 3c) and phase $\phi_{\mathrm{raw},n}$ (Fig. 3a) in complex form:

$$A_{\mathrm{raw},n} = a_n \exp\left(i\phi_{\mathrm{raw},n}\right).$$ (3)

Prior to the FFT the deramped waveform is zero-padded, i.e. lengthened with zeros by a padding factor $p$. This increases the sampling rate of the FFT and reduces the range bin increment to $\Delta R_{\mathrm{bin}} = \delta_R/p$, effectively corresponding to a sinc interpolation of the decomposed amplitude and phase signals. Brennan et al. (2014) suggest using $p \geq 2$ for resolving ambiguities in the phase because otherwise the range bins are separated by more than a full phase cycle (or wavelength) at the center frequency of the pRES. Note, however, that because zero-padding is essentially an interpolation, the intermediate phase cycles cannot be recovered. Nevertheless, the signal phase is typically interpreted relative to an idealized reference phase which, for the center of the $n$-th range bin, is given by

$$\phi_{\mathrm{ref},n} = n\frac{2\pi f_{\mathrm{c}}}{Bp} - n^2\frac{\pi}{Bp^2T}$$ (4)

(Brennan et al., 2014). The second term is negligible over the operation range of the pRES because $f_{\mathrm{c}} \gg 1/T$, so that $\Delta\phi_{\mathrm{ref},\mathrm{bin}} = 2\pi f_{\mathrm{c}}/(Bp)$. Therefore, $p \geq 2$ ensures that the phase variation of the reference phase between range bins is smaller than $2\pi$. In addition, a high padding improves the representation of the reference phase, which is why we apply a padding factor of $p = 8$ for the subsequent LO-SAR processing.

Subtracting the reference phase $\phi_{\mathrm{ref}}$ from the raw signal results in:

$$\begin{aligned} A_{\mathrm{cor},n} &= a_n \exp\left(i\left(\phi_{\mathrm{raw},n} - \phi_{\mathrm{ref},n}\right)\right) \\ &= a_n \exp\left(i\phi_{\mathrm{cor},n}\right), \end{aligned}$$ (5)

where the phase difference between raw and reference phase is denoted as the corrected phase $\phi_{\mathrm{cor}}$.

We note that the corrected phase appears constant along sloped IRHs (Fig. 3b) which we will exploit during the following LO-SAR processing for estimating these slopes. The reason for this is that changes in the reflection range $\Delta R$ are directly

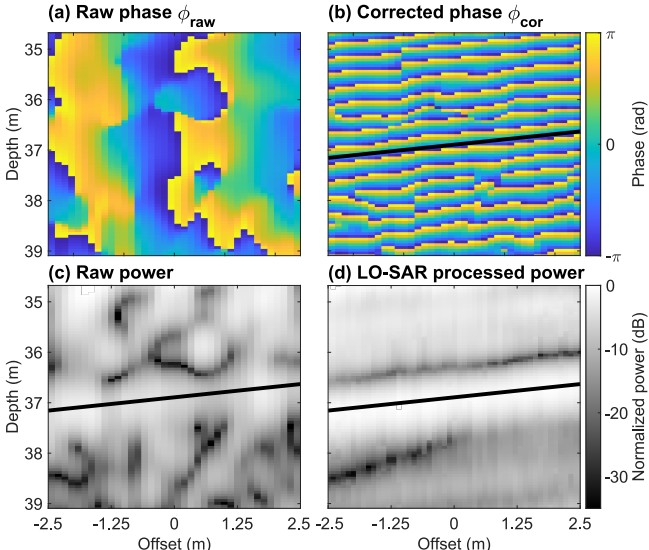

**Figure 3.** An example segment of the pRES profile over one synthetic aperture length. Phase signal before (a) and after (b) correction with the reference signal. Power before (c) and after (d) LO-SAR processing. The slope of the central reflection is indicated by a black line and obtained in (b) by fitting to a coherent section of the reference corrected phase.

related to changes in the raw phase (MacGregor et al., 2015):

$$\Delta\phi_{\text{raw}} = \frac{4\pi\Delta R}{\lambda_{\text{c}}}, \tag{6}$$

where $\lambda_{\text{c}} = c/\left(\sqrt{\varepsilon_r}f_{\text{c}}\right)$ is the wavelength that corresponds to the center frequency. For sloped IRHs, this results in an along-track gradient of the raw phase between adjacent traces $\Delta\phi_{\text{trace}}$ (Fig. 3a) which is related to the IRH slope $S$ by

$$S = \arctan\left(\frac{-\Delta R_{\text{trace}}}{\Delta d}\right) = \arctan\left(\frac{-\lambda_{\text{c}}\Delta\phi_{\text{trace}}}{4\pi\Delta d}\right). \tag{7}$$

As the raw phase is approximately constant in the vertical (Fig. 3a), the horizontal phase gradient due to a change of an IRH range by $\Delta R_{\text{bin}}$ is exactly compensated by subtraction of the reference phase, causing the corrected phase to be constant along IRHs (Fig. 3b).

In this study, we expand the standard FMCW signal processing by cropping the traces at the air wave return and shifting the phase at zero travel time to a uniform value. This accounts for spurious $\pi$-offsets which appear in some of our traces for unknown reasons. Finally, we apply a density correction to account for faster radar wave propagation in firn, which can constitute a significant portion of the thickness of mountain glaciers, whereas the reflection range $\tilde{R}$ as given in Eq. (1) is computed by assuming a constant permittivity of $\varepsilon_{r,\text{ice}}$. To allow for a variable permittivity, $\tilde{R}$ is translated into two-way travel time $\tau = 2\tilde{R}\sqrt{\varepsilon_{r,\text{ice}}}/c$ and then converted back into a density corrected range $R$ by

$$R\left(\tau\right) = \int_{0}^{\tau} \frac{c}{2\sqrt{\varepsilon_r}}\,d\tau', \tag{8}$$

using the relative permittivity $\varepsilon_r$ as given by the Looyenga (1965) mixture model and density data from an ice core. This approach is justified, because the permittivity of ice and air are approximately constant over the frequency range of the pRES (Bohleber et al., 2012).

## 3.2 Layer-optimized SAR processing

Castelletti et al. (2019) introduced LO-SAR processing to improve the detection of inclined IRHs in a radargram. This method aims to enhance the signal strength of IRHs by coherently integrating their backscattered power over a fixed synthetic aperture length $L_{\mathrm{SAR}}$, where the signal-to-noise ratio (SNR) increases proportional to $L_{\mathrm{SAR}}$ (Castelletti et al., 2019). Because according to Eq. (6) the phase varies in the along-track direction for inclined IRHs, it is necessary to correct for this phase gradient before the coherent integration to avoid destructive interference. In Castelletti et al.'s (2019) implementation of LO-SAR processing, for each grid point on the radargram the backscattered power is integrated along its range bin over $L_{\mathrm{SAR}}$ after applying an iteratively found optimal phase shift that maximizes the SNR. Following Eq. (7), this optimal phase shift then provides the local slope of IRHs as a byproduct.

Here, we tailor Castelletti et al.'s LO-SAR implementation to mobile pRES data, where we follow the different approach of first determining the englacial slopes, before using these to perform the coherent integration directly along the IRHs. This has the two advantages that the slopes can be filtered prior to the coherent summation to remove outliers and that the spurious integration of signal power from nearby IRHs is avoided for any grid point on the radargram. A comparison between our approach, the original implementation and moving averaging without correcting for the phase gradient is provided in Fig. S1.

In our implementation, the LO-SAR processed radargram is formed on an equidistant grid where the processed traces are located on a smoothed version of the observed profile line with a spacing of $10\,\mathrm{cm}$. The processing is done grid point by grid point. In the following, we characterize these points $(x, z)$ by their positions $x$ and depths $z$. For every position, all $M$ observed traces within a distance of $L_{\mathrm{SAR}}/2$ are considered in the radargram formation. We use $L_{\mathrm{SAR}} = 5\,\mathrm{m}$ which gives a good improvement in radargram quality, while ensuring that IRHs can be assumed to be linear over the full synthetic aperture length.

The local englacial slopes $S$ at all grid points are determined first. Following MacGregor et al. (2015), these can in theory be inferred from the unwrapped along-track phase gradient as given by Eq. (7). However, this method directly translates uncertainties in the phase, e.g. due to occasional faulty traces and phase jumps at the transition between IRHs, into uncertainties in the slope, which in our data limits the accuracy of slope detection in this way.

We circumvent this problem by not only considering the phase-gradient, but also the spatial coherence of the phase during slope estimation. For this we exploit that the corrected phase is constant along IRHs (Fig. 3b). In practice, we iterate for each grid point over a range of slopes $s$ (from $-30°$ to $30°$ in steps of $0.2°$, a discussion of this range is provided in Section 5.3) to compute the coherence of the corrected phase along a line with length $L_{\mathrm{SAR}}$ and slope $s$, centered around $(x, z)$. Its range at the $M$ observed traces within the synthetic aperture length is given by

$$R_m(x, z, s) = z + (x_m - x)\tan(s), \tag{9}$$

with the position of the observed traces $x_m$. The corrected signal of the $m$-th trace on that line $A_{\mathrm{cor},m}(x,z,s)$ is then obtained by weighting the values in the adjacent range bins above and below $R_m(x,z,s)$. Using these, we define the local phase coherence at $(x,z)$ along slope $s$:

$$
\begin{aligned}
C(x,z,s) &= \left| \frac{1}{M} \sum_{m=1}^{M} \frac{A_{\mathrm{cor},m}(x,z,s)}{|A_{\mathrm{cor},m}(x,z,s)|} \right| \\
&= \left| \frac{1}{M} \sum_{m=1}^{M} \exp\left(i\phi_{\mathrm{cor},m}(x,z,s)\right) \right|.
\end{aligned}
\tag{10}
$$

The slope along which the coherence is largest then gives the local englacial slope $S$:

$$
S(x,z) = \arg\max_s \left( C(x,z,s) \right).
\tag{11}
$$

This approach gives a more consistent slope estimate than the direct computation using Eq. (7), but still may provide erroneous values at the interface between different IRHs. To remove these, we apply a moving median filter to the slope field using a filtering window of $2\,\mathrm{m} \times 2\,\mathrm{m}$.

Finally, the LO-SAR processed radargram is obtained by coherent summation of $A_{\mathrm{cor}}$ along the determined slope $S$:

$$
A_{\mathrm{LO-SAR}}(x,z) = \left| \frac{1}{M} \sum_{m=1}^{M} A_{\mathrm{cor},m}(x,z,S(x,z)) \right|
\tag{12}
$$

This defines the LO-SAR processed amplitude $A_{\mathrm{LO-SAR}}$ at each grid point (Fig. 3d).

## 4 Proof of concept at Colle Gnifetti

The presented mobile pRES setup is tested in a proof of concept study which aims to image the deep englacial stratigraphy of Colle Gnifetti.

### 4.1 Study site

The Colle Gnifetti is located in the Monte Rosa massif (Swiss–Italian Alps) at an altitude of around $4450\,\mathrm{m\,a.s.l.}$ and forms the upper accumulation zone of Grenzgletscher. It is characterized by low accumulation rates due to wind driven snow erosion (Alean et al., 1983), englacial temperatures below $0^\circ\mathrm{C}$ (Hoelzle et al., 2011), and low horizontal flow velocities near the divide (Lüthi and Funk, 2000). In combination, these conditions favor the formation of the longest, still well-preserved, glacial climate record in the European Alps. Several ice cores have been drilled at the site (Fig. 4.), entailing over $1000$ years long climate and environmental records (Bohleber et al., 2018).

This study considers in particular the CG03 and KCC ice cores (Fig. 4). CG03 was drilled in 2003 and analyzed by Sigl et al. (2018a) over its upper $57.2\,\mathrm{m}$. Firn density data of CG03 were obtained by weighting ice core segments of $\sim 70\,\mathrm{cm}$ length (M. Sigl, personal communication; Sigl et al., 2018b). In addition, ion concentrations of Calcium, Sodium, Ammonium, Nitrate and Sulfate ($\mathrm{Ca^{2+}}$, $\mathrm{Na^{+}}$, $\mathrm{NH_4^{+}}$, $\mathrm{NO_3^{-}}$ and $\mathrm{SO_4^{2-}}$) were measured, with the record being extended at the top by the shallow

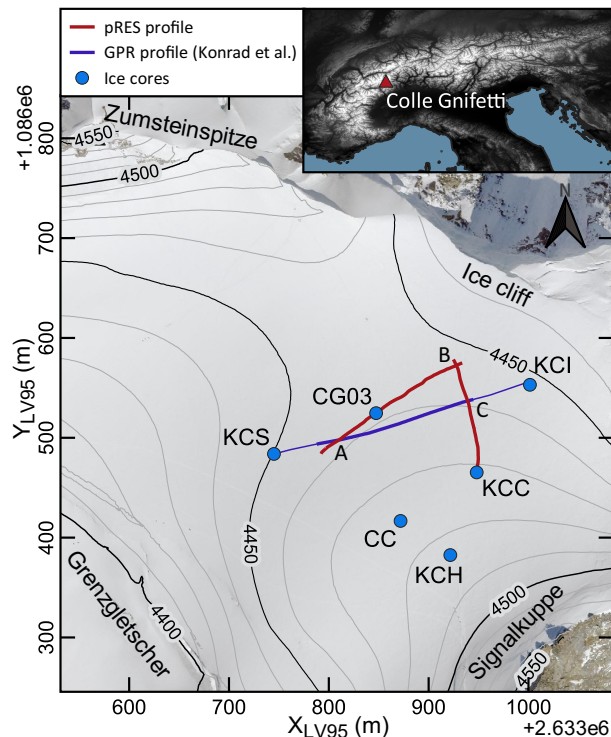

**Figure 4.** Overview of Colle Gnifetti indicating the location of the acquired mobile pRES profiles (red lines). The blue line marks a GPR profile from Konrad et al. (2013) that is used for comparison with the thick part indicating the section shown in Fig. 5a. The intersection points between the radar profiles are labeled (A–C) and blue dots mark ice core drill locations at the time of drilling. The inset shows the location of Colle Gnifetti within the European Alps. Source of orthophoto and local elevation model: Federal Office of Topography swisstopo. Source of global elevation model: NASA Shuttle Radar Topography Mission (2013).

core CG15, drilled at the same location in 2015 (Sigl et al., 2018a, b). Using radiocarbon dating, Jenk et al. (2009) found indications of more than 10 ka old ice at the base of the CG03 ice core, reaching potentially back into the last glacial period. The KCC ice core was drilled in 2013 at the southern flank of Colle Gnifetti. For this core, firn density data were obtained by X-ray computer tomography in $\sim 2\,\mathrm{mm}$ resolution (Freitag et al., 2018). The deep stratigraphy at KCC is moreover interesting
because of a discontinuity in its chronology near the base, perhaps caused by englacial folding (Hoffmann et al., 2018).

### 4.2 Detection of deep englacial stratigraphy

The previous ice core drilling efforts at Colle Gnifetti were supported by radar profiling surveys to reveal the englacial stratigraphic structure and to link the different ice core chronologies (Eisen et al., 2003; Konrad et al., 2013). These surveys used conventional impulse GPR systems, for which the transect between KCS and KCI (Fig. 5a) from Konrad et al. (2013) is a
205 representative example. In these studies, IRHs were mapped in the upper 30–50 % of the ice column, whereas the deeper ice appeared basically echo-free (Eisen et al., 2003). The deepest IRH detectable in the KCS–KCI transect had an age of $78 \pm 12$ a

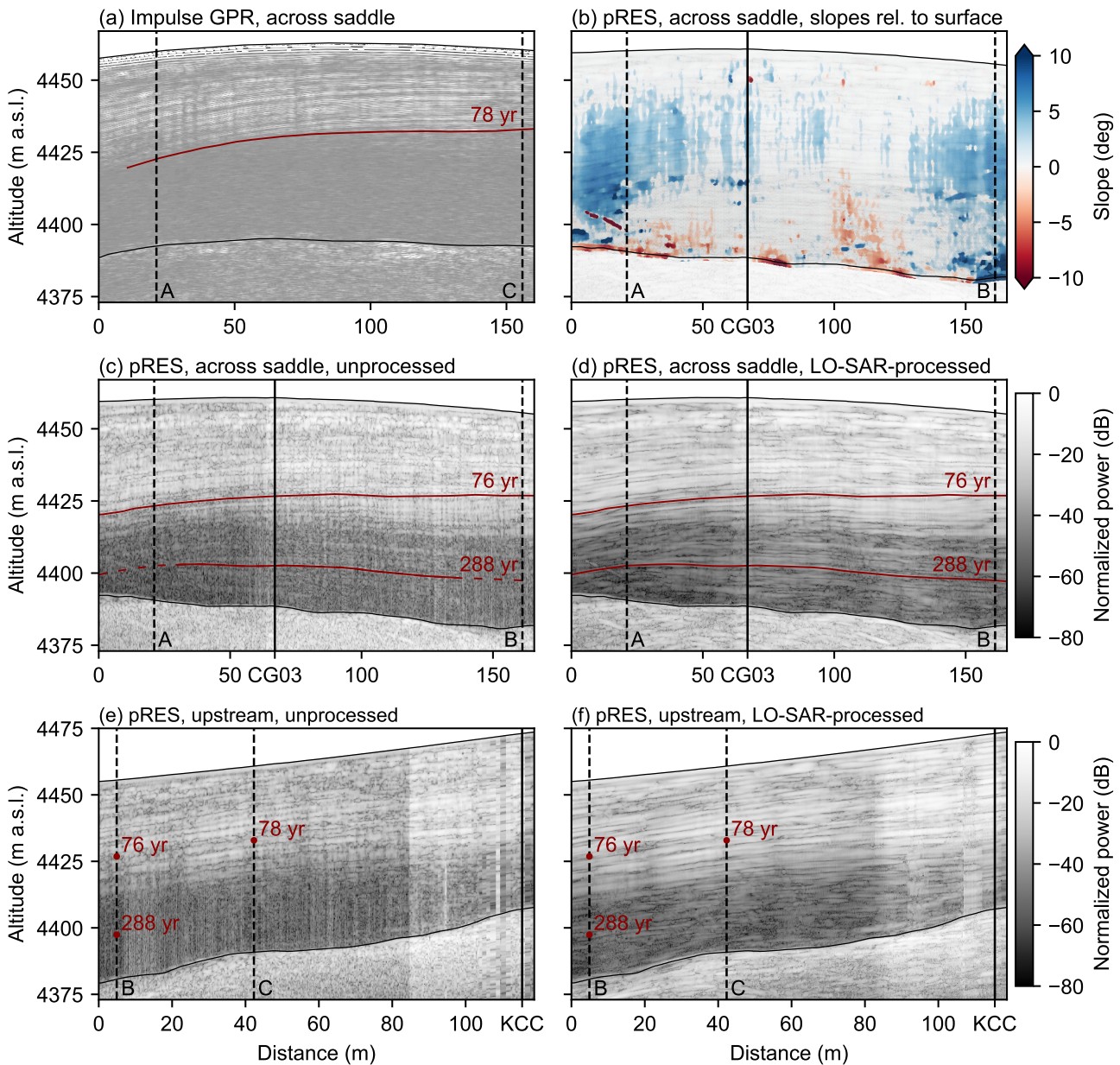

**Figure 5.** Comparison of radargrams (a–d) across Colle Gnifetti and (e–f) upstream towards KCC. (a) Radar data of a impulse GPR (Konrad et al., 2013); (b) IRH slopes relative to the surface derived from the corrected phase of the mobile pRES data; (c, e) mobile pRES data before and (d, f) after LO-SAR processing. The CG03 and KCC ice core locations are marked by vertical black lines and the intersection points between the three profiles are denoted (A–C) and marked by black dashed lines. Red lines mark traced IRHs. In the upstream profile (e–f), the continuous tracing of deep IRHs is prevented by lower data quality and red dots only indicate traced IRHs in the intersecting profiles.

at the time of data acquisition, which corresponds to the year 1930 CE. Note that Konrad et al. (2013) did not assign an age value to this IRH, and we repeated their approach for younger IRHs of averaging the KCS and KCI ages. In these data, the ice–bedrock interface is inconsistently detected and in some cases cannot be identified (e.g., Fig 3.8 in Bohleber, 2011).

In this proof of concept study, we aim to expand the depth range over which IRHs can be detected. In September 2021, we acquired 285 m of phase-coherent radar data using the mobile pRES (Fig. 4). It is divided into a 166 m long profile across the saddle that passes the CG03 ice core and intersects with the KCS–KCI transect by Konrad et al. (2013), and a second 119 m long upstream profile towards KCC/south. Already in the raw mobile pRES data ice and bedrock are clearly distinguishable and deeper IRHs than previously observed are visible (Fig. 5c). In parts these IRHs are yet difficult to trace due to persistent speckle
noise and because their quality deteriorates in the flanks of the saddle where they are more inclined. For these reasons, LO-SAR processing is applied as described in Section 3.2. In this process the IRH inclination relative to the surface is quantified and values of about $10°$ are attained at both ends of the across-saddle profile (Fig. 5b). The LO-SAR processed data reveals deep IRHs which were not apparent in the previous surveys (Figs. 5d and S1).

    In the second profile collected upstream towards KCC (Fig. 5e–f), LO-SAR processing also reveals the presence of deep
specular reflections. However, in this part the quality of our data got diminished by technical problems during data acquisition which might have arisen from overexerting the data writing capacity of the pRES after collecting $> 1000$ individual traces in a single folder. Thereby, various later recorded traces were lost or showed an enhanced noise level. In addition, this profile might have been affected by a less stable positioning of the mobile pRES system at each point due to the increased surface slope. As a consequence, deep IRHs in the upstream profile are disturbed and not continuously traceable, which prevents studying the
potentially complex deep stratigraphy at KCC and reduces the capability of our data to link the CG03 and KCC ice cores.

    For assessing the general potential of the mobile pRES, we therefore focus on the across-saddle profile. In this radargram, we trace two IRHs for comparison with the deepest IRH that was detectable in previous studies (Fig. 5). The first IRH matches the 78 a-horizon in the impulse GPR data with respect to depth at the intersection between both profiles and the second one marks the deepest reflection that is continuously traceable in the LO-SAR processed across-saddle radargram.

The two IRHs are dated by converting their range into units of m water equivalent to link them with the age–depth model of Jenk et al. (2009) for the CG03 ice core that is located on the profile (Fig. 6). Uncertainties in the IRH dating arise both from dating of the ice core and from estimating the reflection range. To obtain ice core dating uncertainties, we computed the confidence limits of the age–depth model at both IRHs giving an uncertainty of 5 a in the upper and 27 a in the lower traced IRH. The range uncertainty is again attributed to two factors: To the inherent uncertainty associated with the width of radar
reflections and to the uncertainty in the firn correction applied to the radar data. The latter was estimated as the given range difference between using the KCC or CG03 firn density data for the density correction, as these two ice cores cover the density variability in our study area. In combination, both factors give a range uncertainty on the order of 1 m (Fig. 6). The full dating uncertainty of both IRHs were computed by error propagation of all three contributing factors.

    In this way, we assigned an age of $76 \pm 7$ a or the year 1945 CE to the upper IRH, that matches the deepest detectable
horizon ($78 \pm 12$ a) in the survey by Konrad et al. (2013). The lower traced IRH was dated to an age of $288 \pm 35$ a (i.e. the year 1733 CE). As it represents the deepest continuously traceable IRH in our data, it demonstrates that LO-SAR processed

mobile pRES data can be used to inter-compare ice-core chronologies at Colle Gnifetti for larger depth and age ranges than what was possible before. The system can in principle illuminate the englacial structure over the whole ice column. Even below the 288 a-horizon IRHs are visible in parts in our data, which implies that older IRHs are potentially also preserved and might be traceable by further improvements in the acquisition and processing.

In summary, the data collected with the mobile pRES reproduces and expands the results of previous surveys. This can be attributed to a combination of accurate data acquisition and system hardware leading to a high SNR, and to the enhancement of signal coherence by the applied LO-SAR processing.

A caveat in interpreting deep IRHs and linking them to ice cores lies in their associated depth-uncertainty and in potential interference by the bedrock reflector. At the location of the CG03 ice core, the ice–bed interface appears in our radar data at a range of 72 m (Fig. 6d), whereas the ice core itself has a length of 80.2 m (Jenk et al., 2009). This mismatch can be attributed to a potentially non-vertical orientation of the borehole, and steep gradients in the bedrock topography of Colle Gnifetti, which affect the apparent position of the bedrock reflector in the radar data (Eisen et al., 2003; Bohleber, 2011). Following Moran et al. (2000), three-dimensional array processing and correspondingly dense profile grids are needed at steeply sloping topography to accurately measure the bedrock reflector depth. With only two-dimensional profiling, an accuracy on the order of 10 m is to be expected. Moreover, unlike traditional SAR processing methods, the LO-SAR processing does not provide a range migration, but merely aims at improving the SNR of specular reflections in the radargram.

Because of the complex basal topography, it is moreover possible that near-basal IRHs are masked by the 15 dB stronger cross-track bedrock reflections, if these IRHs are not similarly recorded from cross-track angles. Expanding our setup with a multi-channel cross-track antenna array (Castelletti et al., 2017; Holschuh et al., 2020; Scanlan et al., 2020) would allow to decipher the true origin of the near-basal IRH and bed returns and help in locating, separating and interpreting their signatures.

### 4.3 Origin of reduced radar backscatter

By observing deep IRHs that are undisturbed and nearly flat, it can be excluded that their apparent absence in earlier surveys was caused by buckling or folding of reflectors, or a complete degradation of dielectric contrasts. Yet, a distinct transition in radio-backscatter between the upper and the lower part of the ice column is also present in the LO-SAR processed pRES data. The pRES power profile at CG03 (Fig. 6d) shows a highly reflective regime in the top 36 m, where reflections mostly attain values above $-25$ dB, and a low reflective regime below 45 m depth, where the returned power rarely exceeds $-40$ dB. In between these two regimes, the power level drops abruptly over less than 10 m. The bedrock reflections have a normalized power above $-25$ dB. Variations of density and acidity, together with COF, are reflection mechanisms that can form IRHs and control the reflectivity. Here, we investigate if the dominant reflection mechanisms at Colle Gnifetti and the origin of the drop in backscattered power can be identified by direct comparison of the reflected power of observable IRHs with density and acidity records from ice cores.

Both at CG03 and KCC, firn compaction is mostly confined to the upper $\sim 40$ m (Fig. 6a). Strong density variations occur particularly in the top 15 m, where melt layers can be identified in the high-resolution record of the KCC core (Freitag et al., 2018). These variations are likely causing near-surface IRHs. Further down, especially below 35 m depth, density contrasts are

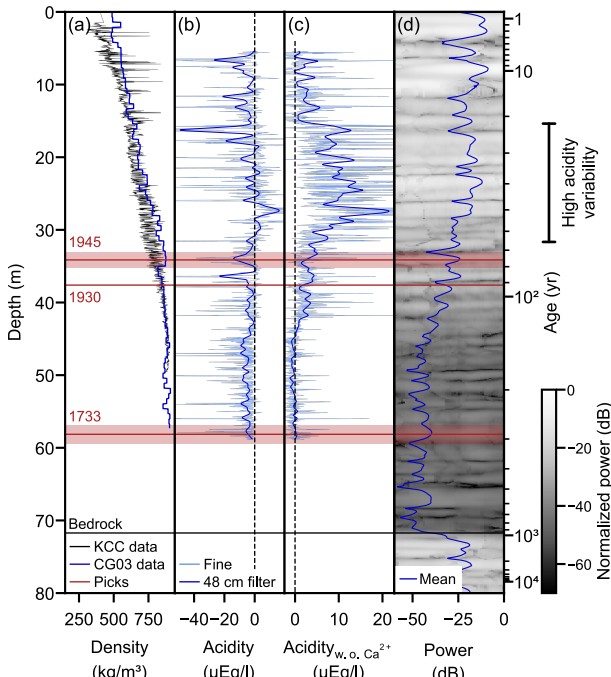

**Figure 6.** Comparison of mobile pRES power with data from the CG03/CG15 (Sigl et al., 2018b) and KCC (Freitag et al., 2018) ice cores. (a) Firn density; (b) Acidity estimated by ion balance in fine resolution (2 cm) and after applying a 48 cm Gaussian filter; (c) as (b), after excluding $Ca^{2+}$ from the ion balance; (d) Radar section (2 m) around CG03 and its mean power. Red lines mark traced IRHs, with error bands of the 1945- and 1733-horizons illustrating the uncertainty in the range estimate. Ages are computed using the age–depth model of Jenk et al. (2009).

significantly weaker and show no correlation with the backscattered power. The reflectivity at depth is therefore not controlled by density.

Accordingly, deep IRHs must be caused by impurities which determine the acidity and perhaps may induce localized gradients in the COF (Drews et al., 2012). By computing the ion balance between the $Ca^{2+}$, $Na^+$, $NH_4^+$, $NO_3^-$ and $SO_4^{2-}$ con-
centrations measured in the CG03/CG15 ice cores (Sigl et al., 2018a, b), we obtained an estimate of the acidity with $\sim 3$ cm resolution (Fig. 6b). In addition, for comparison with the pRES signal we smoothed this estimate using a Gaussian filter with a width of 48 cm which corresponds to the mean resolution of the pRES data in firn.

The ion balance is dominated by alkaline peaks that are induced by $Ca^{2+}$ which is a proxy for Saharan dust. An example for a major Saharan dust deposition event is the 1936 CE peak in 36.5 m depth (Sigl et al., 2018a) that appears as a strong IRH
in the radar data (Fig. 6d) and marks the onset of the power drop. Major Saharan dust deposition events occur on a decadal timescales, with an increasing frequency only in the last 4 decades, attributed to increasing drought conditions in northern Africa (Sigl et al., 2018a). In combination with other natural deposition, for example due to volcanic activity and forest fires, this sets the natural background for acidity variations at Colle Gnifetti.

By excluding $Ca^{2+}$ from the ion balance, a trend from alkaline conditions in the pre-industrial to more acidic conditions during the 20th century becomes apparent (Fig. 6c). This trend initiates at a depth of $45\,m$, which corresponds to the year 1875 CE and the onset of industrialization in Western Europe (Sigl et al., 2018a). Accordingly, the transition in acidity can be attributed to an increase in anthropogenic pollution. In particular, the deposition of $NH_4^+$, $NO_3^-$ and $SO_4^{2-}$ is largely controlled by anthropogenic activity (Preunkert et al., 2003; Schwikowski et al., 1999) and correlates with other industrial emissions (e.g., refractory black carbon, Sigl et al., 2018a). Eisen et al. (2003) suggested that this acidity shift induces the change in reflectivity at Colle Gnifetti. A similar contrast in reflectivity at the glacial-interglacial transition is present across the Greenland Ice Sheet and is also thought to be induced by a transition in the amount of impurity deposition (Karlsson et al., 2013).

The more acidic conditions in the upper ice column are accompanied by a generally higher variability of the acidity, which is most pronounced in the fine resolution record between $15$–$32\,m$ depth. This high acidity variability strongly correlates with the strength of backscattered power. We conclude that it is not the general acidity level itself which induces the shift from a low- to a highly-reflective regime at Colle Gnifetti, but that it is caused by the associated higher rate at which deposition of anthropogenic pollution occurs.

In addition to altering the acidity, impurity layers might control the grain size (Kerch, 2016) and thereby induce localized changes of the COF (Drews et al., 2012). Potentially, this mechanism can additionally modulate the reflectivity at Colle Gnifetti. In a seismic profile collected across the KCI ice core, Diez et al. (2013) detected an englacial reflection $5\,m$ above the bedrock, which they attributed to changes in the crystal orientation. In our data, this reflection might correspond to the comparatively strong IRH at $67\,m$ depth.

## 5 Discussion

### 5.1 Feasibility assessment

Our proof of concept study demonstrates the capability of the pRES to expand the range over which englacial stratigraphy can be detected which is a clear improvement over previous surveys that used impulse GPRs. However, it comes at the cost of significantly slower data acquisition (Section 2.1) that takes several hours as opposed to several tens of minutes with impulse systems. This results from the required high spatial sampling rate and the long chirp time of 1 s of the pRES (which is not a limiting factor for its intended stationary surveys). In combination, these factors make data collection in stop-and-go mode necessary.

This raises the question whether impulse GPRs can achieve similar detection ranges when being operated in stop-and-go mode and by applying high stacking rates, equivalent to the 1 s chirp time, to suppress incoherent backscatter. Although we cannot answer this question directly, we note that even if these GPRs can achieve a SNR comparative to the unprocessed pRES data (Fig. 5c), they will not be able to match the SNR potentially achievable by phase-coherent radar processing (Fig. 5d). We suggest that future research should not focus on improving imaging with impulse GPRs by making data acquisition more painstaking but rather aim at accelerating the profiling capabilities of FMCW radars. The laborious data acquisition is the key

limitation for mobile operation of the pRES and restricts the current applicability of the system to targeted small scale surveys. In the following, we discuss how to accelerate data acquisition in future deployments.

The required spatial sampling rate for LO-SAR processing is set by the fact that reflections need to be traceable (Schroeder et al., 2019). The slope of IRHs can only be derived from the phase when it changes by less than $\pi$ between neighboring traces, otherwise, spatial aliasing occurs. Thus, following Eq. (7) the minimal required trace spacing to resolve IRHs with a slope $s$ is given by

$$\Delta d < \frac{\lambda_{\mathrm{c}}}{4\tan(s)}. \tag{13}$$

In snow and firn-covered glacial ice, GPRs can only record IRHs with a maximum slope of approximately $45°$ (Holschuh et al., 2014). Accordingly, a trace spacing of less than $14\,\mathrm{cm}$ is required to record IRHs of such slopes. This also suffices the threshold of $\Delta d < \lambda_{\mathrm{c}}/2$ to avoid the occurrence of grating lobes when imaging (basal) point reflectors (Kapai et al., 2022). A coarser trace spacing can be used, when only IRHs with a lower inclination are expected at the study site.

Nevertheless, if the requirement for stop-and-go measurements can be relaxed, a high spatial sampling rate is less of a problem. This can either be achieved by reducing the chirp time or by reducing the potential noise due to movement of the pRES during chirp acquisition. Electrical noise by cable motion can be reduced by using phase-stable cables (Hati et al., 2009). However, increased noise by averaging signals of multiple reflections and the occurrence of Doppler blurring and grating lobes (Kapai et al., 2022) can only be avoided with the pRES when moving at very low speeds. Fortunately, these problems are not of physical, but of technical nature and can again be avoided by reducing the chirp time as discussed in the following.

## 5.2 System improvements

The chirp time $T$ of a FMCW radar is set by its hardware design and operation range. This is because the deramped frequency at a given range is determined by $T$ (Eq. 1). Thus, the sampling rate of a FMCW radar system's data logger sets a lower limit on $T$. The pRES, for example, was designed to monitor up to $2\,\mathrm{km}$ thick polar ice, which for a chirp time of $1\,\mathrm{s}$ gives a maximum deramped frequency of $4.7\,\mathrm{kHz}$. The build-in data logger with an effective sampling rate of $40\,\mathrm{kHz}$ was selected accordingly (Brennan et al., 2014). For deploying the pRES on thinner ice, the chirp time can be reduced. For example, for surveying less than $200\,\mathrm{m}$ thick ice, as we do in this study, a $90\,\%$ shorter chirp time could safely be set without exceeding the sampling rate of the data logger. Yet, this reduction would also result in an increased noise level which might need to be compensated by stacking. To completely avoid the stop-and-go requirement, a redesigned FMCW system is desirable that is similar to many airborne systems whose chirps are much shorter ($\leq 10\,\mathrm{\mu s}$).

A second limitation of the pRES lies in its range resolution, which is 2.5 times coarser compared to the previously used $250\,\mathrm{MHz}$ impulse GPRs. Following Eq. (2), the vertical resolution of FMCW radars is determined by their bandwidth: $\delta_{R,\mathrm{FMCW}} = c/\left(2\sqrt{\varepsilon_r}B_{\mathrm{FMCW}}\right)$. For the pRES with $B = 200\,\mathrm{MHz}$ this results in a vertical resolution of $0.43\,\mathrm{m}$. The resolution of GPRs is often approximated by $\delta_{R,\mathrm{GPR}} = \lambda_{\mathrm{GPR}}/4 = c/\left(4\sqrt{\varepsilon_r}f_{\mathrm{GPR}}\right)$, giving a value of $0.17\,\mathrm{m}$ for $250\,\mathrm{MHz}$ impulse GPRs. These radars are consequently better at detecting shallower IRHs. To achieve a comparable resolution with a redesigned FMCW system, a bandwidth of $500\,\mathrm{MHz}$ would be required.

Besides these potential improvements in the FMCW radar design, certain modifications to our mobile pRES setup should be considered. Here, we used a HH antenna orientation in which the E-plane matches the imaging plane. In this configuration, the radiation pattern of the skeleton slot antennas extends further along-track than cross-track and ground targets are illuminated over more traces, however, the direct wave between the antennas is also stronger which can promote signal clipping (Vaňková et al., 2020). To suppress the direct wave, a perpendicular orientation of the E-plane to the profiling direction (VV orientation) might be advantageous. Note that these characteristics are antenna-specific and do not generally apply. In addition, to follow the low-cost approach of the pRES, we suggest to combine it with low-cost GNSS receivers that can achieve a positioning accuracy that is comparable to commercial instrumentation (Still et al., 2023; Pickell and Hawley, 2024).

### 5.3 Slope estimation

The presented LO-SAR processing is also computationally expensive, primarily because of the slope estimation algorithm. For short profiles, this is less of a problem (in the examples provided here the compute time was 12 hours on a normal performing desktop computer), but it might be a restriction for more extensive mobile pRES surveys.

Direct computation of englacial slopes from horizontal gradients of the phase, as introduced by MacGregor et al. (2015), is significantly faster. But it also gives a less accurate estimate, because it operates on a point-by-point (i.e. zero-dimensional) base and directly translates noise in the phase into noise in the slopes. For this reason, we derive the slopes of IRHs by matching slope lines to the corrected phase. As this is a one-dimensional approach we combine information from adjacent traces, which entails some horizontal averaging and thereby provides a more accurate slope estimate.

A third method to derive englacial slopes is based on the Radon transform which operates on a two-dimensional window and computes line integrals over a range of possible slopes and intercepts (Holschuh et al., 2017). Similar to our approach, the Radon transform is computed for every grid point and the correct slope is selected as the one which gives the most coherent stack. Although this is even more computationally expensive, it does not improve the slope estimate further compared to our approach. This can be attributed to the fact that we essentially perform a one-dimensional Radon transform which only considers an intercept of zero. This is possible because we obtain the coherence along the line integrals from the coherence of the phase and not from the coherence of structural patterns, as the two-dimensional Radon transform does.

The computational costs of our slope estimation directly correlate with the length of the slope iteration range. The selection of the iteration range is based on two factors. Its boundaries must cover the range of slopes that are to be expected, and the step size needs to be sufficiently small to ensure that a mismatch does not result in destructive interference along the slope line. The limit for the occurrence of destructive interference can be estimated from the phase offset between range bins of $3\pi/p$ (Eq. 4). At a phase mismatch of $\pi$ between the end points of the slope line, these will be completely out of phase, which for $p = 8$ equates to a mismatch of $8/3$ range bins. For an aperture length of $5\,\mathrm{m}$ this corresponds to a slope error of $\sim 2°$. As the compute time was not a limiting factor in our proof of concept, we choose a wide slope range of $-30°$ to $30°$ in steps of $0.2°$ to not a priori prevent the detection of potential stronger inclined reflections in the near-basal ice and to sufficiently avoid the destructive interference limit of $2°$.

Our final slope estimate shows some noticeable vertical bands (Fig. 5b). These are most likely not englacial signals, but are caused by antenna tilt due to small-scale surface undulations. However, the slopes are only an intermediate product in our LO-SAR processing routine. In the subsequent coherent summation along IRHs, we specifically aim to follow IRHs, also across such undulations. For this reason, these artifacts are of no concern in our study, but might be relevant in other applications of the slope estimate.

Finally, it should be noted that our use of the corrected phase during slope estimation contradicts with its intended purpose of providing a fine range offsets for each range bin, as suggested by Brennan et al. (2014). They introduced it as the phase difference between the raw and the reference phase, which following Eq. (6) would translate into a fine range offset to measure the absolute range of a reflection in a Vernier-like process (Brennan et al., 2014). However, this approach relies on the assumption that the raw phase values in each range bin are independent. This would be expressed by a clear inter-bin variability of the raw phase, which cannot be observed (Fig. 3a). Instead, strong reflections spread out over several range bins, across which the raw phase tends to be stable and distinct jumps of the raw phase only occur at the transition between different reflections. Therefore, the phase of a reflection is only representative for a specific point within its width, potentially its center, which needs to be taken into account for measuring the absolute range of a reflection. The interferometric measurement of the relative change of a reflection range between two measurements is not affected by its inter-bin spread and can be directly computed from phase changes at any range bin.

### 5.4 Future applications

The pRES is used in a broad range of glaciological applications. For example, by performing repeat visits at fixed locations, the pRES can measure ocean-induced melting at the base of ice shelves (Nicholls et al., 2015; Vaňková et al., 2021; Zeising et al., 2022), englacial vertical ice deformation (Gillet-Chaulet et al., 2011; Kingslake et al., 2014) and firn compaction (Case and Kingslake, 2022). With polarimetric techniques that screen the ice with differently polarized electromagnetic waves, the COF can be inferred (Jordan et al., 2020; Ershadi et al., 2022). Our mobile pRES setup expands this list by the enhanced detection of englacial stratigraphy, but also offers the potential to perform the above-mentioned survey types with larger spatial coverage.

Polarimetric pRES measurements can efficiently be conducted with multiple-input multiple-output (MIMO) antenna configurations in which the two transmitting and two receiving polarized radar antennas are orthogonally oriented, respectively. A mobile acquisition of COF data by towing a polarimetric MIMO setup with an autonomous rover was presented by Ershadi et al. (2024). MIMO setups can moreover be used for cross-track signal detection (Castelletti et al., 2017; Holschuh et al., 2020; Scanlan et al., 2020) or for density inversions (Arthern et al., 2013). However, the additional antenna weight restricts the extension of the here presented mobile pRES setup to MIMO configurations, because depending on the surface topography and snow conditions the system would become to heavy for hand-towed operation. For obtaining a lightweight FMCW MIMO system, the use of an FMCW radar that operates in a higher frequency range and, thus, with smaller and lighter antennas would be required.

For repeat-track surveys, for example to measure the spatio-temporal variability of firn compaction (Medley et al., 2015), no adaptations to our mobile pRES setup are needed. But the correlation of pRES repeat measurements to detect the vertical

displacement of englacial reflections or the ice-ocean interface is thought to rely strongly on accurate repositioning of the system. We therefore suggest that future studies investigate which accuracy in the spatial and directional repositioning is required to develop corresponding acquisition and processing techniques.

## 6 Conclusions

We present a phase-coherent ground-based radar system with low weight and power requirements by mobilizing the pRES. The setup facilitates coherent radar profiling in glaciated areas with low accessibility, as for example on remote mountain glaciers. A centimeter scale positioning accuracy at decimeter trace spacing is achieved by integration of a RTK GNSS. In combination, the coherence and positional accuracy enable the application of SAR processing techniques for enhancing the quality of the obtained radargrams. Here, we specifically focus on improving the imaging of the englacial stratigraphy by tailoring the LO-SAR processing technique by Castelletti et al. (2019) to mobile pRES data. In this process, we demonstrate that after subtraction of the reference phase the corrected phase signal is constant along IRHs. We exploit this characteristic in our implementation of LO-SAR processing by first detecting the slopes of IRHs to filter these before performing the coherent integration of power directly along IRHs.

The capability of the mobile pRES system and LO-SAR implementation are demonstrated in a proof of concept study for detecting the deep englacial stratigraphy of Colle Gnifetti. Previous GPR surveys at the site could detect IRHs only in the upper 30–50 % of the ice column, whereas with our approach the presence of a layered stratigraphy can essentially be detected down to the bedrock. Thereby, an IRH with an age of $288 \pm 35$ a could be traced continuously, which is a clear step forward compared to the detection of a $78 \pm 12$ a old IRH achieved by previous surveys. By comparison to ice core data, we in addition identified that the presence of a highly reflective regime in the upper part of Colle Gnifetti (detectable in previous surveys) compared to a less reflective regime in the lower part (which previously appeared echo-free) is caused by acidic impurity layers that are deposited at a high rate since the onset of industrialization. These improved imaging capabilities of the mobile pRES highlight the possibility to extensively map the deep stratigraphy of Colle Gnifetti and other mountain glaciers in the future. However, our study also demonstrates that the main limitation of the mobile pRES lies in the time-intensity of data acquisition due to its 1 s chirp time. We therefore encourage the development of faster ground-based FMCW radar systems suitable for profiling. Repeat-track surveys are a promising application for the mobile pRES as they require no hardware adaptations. In addition to ongoing efforts of expanding the repeat-track capabilities of airborne chirp systems, this opens the avenue to detect englacial deformation, firn compaction and basal melt rates with the pRES with large spatial coverage and fine resolution.

*Code and data availability.* The raw and processed mobile pRES data are available at https://doi.org/10.1594/PANGAEA.965199 (Oraschewski et al., 2024a) and https://doi.org/10.1594/PANGAEA.965194 (Oraschewski et al., 2024b). The LO-SAR processing code is available at https://doi.org/10.5281/zenodo.12656458 (Oraschewski, 2024).

*Author contributions.* FMO led the data acquisition, analysis and writing of the paper. FMO, RD and OE designed the study outline and FMO, RD, IK and MRE conducted the data acquisition. JH supported the analysis of the data and designed Fig. 1a. All authors contributed to the writing of the final paper.

*Competing interests.* RD and OE are members of the editorial board of The Cryosphere. The authors have no other competing interests to declare.

*Acknowledgements.* We acknowledge the support of Falk M. Oraschewski through a doctoral scholarship provided by the German Academic Scholarship Foundation. Reinhard Drews, Inka Koch and M. Reza Ershadi were supported by Deutsche Forschungsgemeinschaft (DFG) Emmy Noether grant (grant no. DR 822/3-1) and Jonathan Hawkins by the Royal Society Enhancement Award (grant no. RGF\EA\180173), awarded to Dr. Lai Bun Lok. We further acknowledge support from the Open Access Publication Fund of the University of Tübingen.

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
