# Peer review of "Layer-optimized SAR processing with a mobile phase-sensitive radar: a proof of concept for detecting the deep englacial stratigraphy of Colle Gnifetti, Switzerland/Italy"

_EGUsphere, 2023_

## Referee Comment (RC2)

Review of:
**Layer-optimized SAR processing with a mobile phase-sensitive radar for detecting the deep englacial stratigraphy of Colle Gnifetti, Switzerland/Italy**

Submitted to: *The Cryosphere*
Reviewer: Benjamin Hills

**Summary.**

   Oraschewski et al. describe a novel setup and implementation of the popular ApRES instrument, specifically for SAR profiling in alpine environments. Overall, I believe that this work could be highly impactful, particularly for alpine glaciology studies. As I describe in detail below, I believe that some additions to the engineering discussion as well as some organizational changes could elevate the impact of this article. It is well written and a great fit for *The Cryosphere* after these minor revisions.

**General Comments.**

I love this, neat design and I learned a lot. My largest high-level comment is that the article, as written, feels like it is in between a science narrative and an engineering narrative. I believe that you are intending for this to be read as an engineering article with a scientific proof of concept, and I think that some very minor organizational changes could help make that clearer. I would completely segregate the engineering language from the site-specific scientific proof of concept, preferably loading the engineering up front. I would also generalize the engineering discussion as I describe in paragraphs below. Ideally, anyone who is thinking about using a pRES to do profiling in future studies will be coming back to this article to figure out how to optimize their setup.

One plausible organization scheme could be:
1. Introduction
2. Hardware design & Data Acquisition
     - Continuous acquisition vs stop-and-go
     - MIMO (might be too much for this article?)
3. Data Processing
     - Standard FMCW
     - LoSAR
     - Multipass (might be too much for this article?)
     - Polarimetry (might be too much for this article?)
4. Proof of Concept at Colle Gnifetti
     - Site Description
     - Radar Dataset
     - Comparison to ice core (I would put *all* the chemistry background, results, and discussion in this one section, it is interesting but not the focus of your manuscript and confusing when it comes up in three separate places)

5. Discussion
    - Mainly covers the nuance of your proof of concept (which is what you are already doing with the Discussion).
6. Conclusion

As you see in my outline above, I added a few sections which you do not currently cover (e.g. MIMO, multipass, and polarimetry, I expand on each in the following paragraphs). Here, I am encouraging you to consider broader use cases for this instrument rather than limiting the conversation to what you end up doing in the example, that way you can have a bit broader impact with this article. You could also consider adding a section (possible between Discussion and Conclusion?) on "Future Directions" where you would put some of this multipass and polarimetry language.
Sections 2 and 3 obviously have a lot of overlap and may need some consideration in how they are folded together or not.

*ApRES specificity*
I believe you are currently too specific to particular ApRES settings. In my opinion, the engineering language would be more useful if you generalized the numbers for center frequency, bandwidth, and chirp time (all of which can be changed in software as far as I know). Then, in your proof of concept section you give the specifics for that particular case and use those specifics as justification for your survey design (e.g. the stop and go surveying mode).

*MIMO and Polarimetry*
I know that this team has already been thinking about multi-input-multi-output designs, so I am eager to know what they think that might look like for this alpine setting? Is that infeasible here? Too heavy to carry by ski? My tendency would be to include this, even if it is presented as a possible next direction, then go more specific to your single-channel design in the proof-of-concept section.
If you include MIMO in the hardware section you would want to at least mention how the multiple channels helps you, whether that be variable offset (for velocity inversions) or mult-polarization for polarimetry.

*Multipass*
This is the other obvious use case for your design which would not need any hardware changes. If you left out MIIMO and polarimetry that seems fine to me, but I think this should be included because in my mind it is an obvious use case for your instrument. Specifically, what I mean is resurveying the same profile between two different times to measure vertical velocities.

**Specific Comments.**

Title – I would say it is a proof-of-concept at Colle Gnifetti. The current title makes it sound like this instrument is only useful at one site.

Abstract – I believe that "(pRES)" and "(LO-SAR)" don't need the acronym in parentheses since you only use them once in the abstract and you redefine in the text.

L10 – "down to the base…" you could be more specific. "Improved from 50% depth with pulsed radar to 80% depth with lo-sar"?
*Coming back to this since in the discussion you mention the ages that you resolve (78 yr vs 288 yr) I think those are useful numbers to have in the abstrace if it is something you want to emphasize.*

L25 – Instead of the in press article, there are many alternatives you could cite (Arcone et al., 2005; Medley et al., 2013).

L34 – "neither of which are applicable…"

L57 – Instead of "invisible" I might say "unresolved"

L85 – I recently talked with Keith Nicholls about doing pRES profiling and he suggested that I shorten the chirp time. I didn't have a lot of luck with it, but my experience has me thinking about whether you want to change how this gets presented here. If you give some guidelines for how short a chirp you need to profile continuously the community may come back to this paper a lot more.
*I am now seeing that you do talk through some of this in the discussion, but I think talking about it when it is introduced will be helpful (I am guessing that a lot of readers will be primarily interested in your hardware design and processing flow).*

L102-103 – as I said before, it might be more broadly useful to the community to generalize some of the numbers here until you get to your proof-of-concept where you would give specifics.

L113-115 – I might add 'volumetric scattering' in line with my comment above on adding another adjustment.

L134-139 – This is fine to do here in my opinion, but I think it is not precisely correct since you are compressing information over the full bandwidth to get the range in eq. 1. I might just add a bit more nuance so people don't misuse this approximation.

L143-144 – I may be confused the "explicit sum", but I believe Castelletti et al. (2019) did something similar if not the same. They do a coherent sum in their eq. 3 and then "incoherent

averaging" as described in the subsequent paragraph. Maybe you are doing something different; if so, more explanation would be helpful.

L153-154 – This is interesting. I have actually had a lot of luck with the method from MacGregor et al. (2015) (it is much faster as you say), but that has been with MCoRDS data and I hadn't considered the phase uncertainty in this way. I am interested in seeing a comparison figure although I get that most readers are not me and this is not the emphasis of the article, so disregard unless you think it is useful.

L159-176 – I believe this is mostly the same as Castelletti et al. (2019), correct? Either way, I think it is important for you to write it out here (especially since the way you write it is a bit different). Still might be worth citing them and pointing out explicitly where you are (or are not) different.

L181 – comma feels like it is in a weird spot to me

L212 – I believe more details on "poorer data quality" could be exceedingly useful for future studies trying to do this type of thing.

L214-215 – save mention of the ice core for the next section

L230-231 – I believe this result is correct. You might add a citation to Greenland radar work and the Holocene-glacial contrast, longer time period but similar idea.

L248-249 – Nice. "should not be a more painstaking data collection using pulsed GPRs but rather aim at accelerating the profiling capabilities of FMCW radars". Very much agree with this.

L 274-278 – What is required to change the frequency band. I think you can change it in software, but I am guessing the skeleton slots are specifically designed for this band. Could someone design new antennas and use the same unit? As Jonathan has been doing for the HF ApRES but now to a higher frequency band? Could be some interesting additional notes here.

L307 – most likely "not"

L328-335 – Do you think there is also a lot of uncertainty in the density which causes thickness uncertainty? Or you are decently confident on the density?

L336 – I would add that it improves SNR specifically for specular reflectors, which the bed is not in most cases.

L349 – "promote" changes in the COF? Or "caused by" changes in the COF?

L368 – This group could probably convince me (so please do if I am wrong!) but I don't think it is the impurity layers which are causing changes in the COF. Rather that boundaries between two different COFs can cause a reflection.

Conclusion – Too focused on your site. I believe that this radar design will be highly useful to many groups all over the world, do not undersell yourself by making the reader thing it is only for this site!

L388 – Excited to see the processing scripts when you have them up! Please do be sure to post them.

**Figures.**

Figure 1. I am not very familiar with this area so an inset map showing where you are located in the alps and then where on the glacier, those would be helpful. As with my *general comments,* I think you should move this figure lower. Having Figure 1 be a map makes it feel like this is a scientific paper, whereas if you want it to feel more like an engineering paper starting with Figures 2 and 3 would be helpful.

Figure 3. If you decide to include language on MIMO and multipass then I would add more to this flowchart as well.
I was at first confused about the arrow labeled "power and phase of traces in synthetic aperture length". I didn't understand the direction. I would put little arrowheads along that line to help a reader out.

Figure 4. It is tough to see the layers with your red annotation lines over top of them. Either drop the lines and label with points/arrows (as you do in e and f) or make additional panels for interpretation as in Welch & Jacobel (2005) Figures 3a and 5b.

**References**

Arcone, S. A., Spikes, V. B., & Hamilton, G. S. (2005). Stratigraphic variation within polar firn caused by differential accumulation and ice flow: Interpretation of a 400 MHz short-pulse radar profile from West Antarctica. *Journal of Glaciology*, *51*(174), 407–422. https://doi.org/10.3189/172756505781829151

Castelletti, D., Schroeder, D. M., Mantelli, E., & Hilger, A. (2019). Layer optimized SAR processing and slope estimation in radar sounder data. *Journal of Glaciology*, *65*(254), 983–988. https://doi.org/10.1017/jog.2019.72

MacGregor, J. A., Fahnestock, M. A., Catania, G. A., Paden, J. D., Gogineni, S. P., Young, S. K., Rybarski, S. C., Mabrey, A. N., Wagman, B. M., & Morlighem, M. (2015). Radiostratigraphy and age structure of the Greenland Ice Sheet. *Journal of Geophysical Research: Earth Surface*, *120*, 212–241. https://doi.org/10.1002/2014JF003215.Received

Medley, B., Joughin, I., Das, S. B., Steig, E. J., Conway, H., Gogineni, S., Criscitiello, A. S., McConnell, J. R., Smith, B. E., Van Den Broeke, M. R., Lenaerts, J. T. M., Bromwich, D. H., & Nicolas, J. P. (2013). Airborne-radar and ice-core observations of annual snow accumulation over Thwaites Glacier, West Antarctica confirm the spatiotemporal variability of global and regional atmospheric models. *Geophysical Research Letters*, *40*(14), 3649–3654. https://doi.org/10.1002/grl.50706

Welch, B. C., & Jacobel, R. W. (2005). Bedrock topography and wind erosion sites in East Antarctica: Observations from the 2002 US-ITASE traverse. *Annals of Glaciology*, *41*, 92–96. https://doi.org/10.3189/172756405781813258

---

## Author Comment (AC1)

**Response to referee #1 on egusphere-2023-2731**

**General comments:**

This paper presents interesting results from a field survey of mountain glaciers at Colle Gnifetti using a phase-sensitive radio echo-sounder (pRES). The authors obtained the results with a layer-optimized SAR processing method, which is special in that it performs coherent summation over the synthetic aperture along an optimally estimated linear slope. The echograms from pRES revealed deeper internal reflection horizons (IRHs) that were not visible in the results from previous surveys with pulsed GPR. This data set is thus valuable for studying snow accumulation, ice flow and ice core chronologies in this area. The paper also provides valuable insights into different reflection mechanisms of the glacial IRHs at different depths by comparing radar data and ice core records. The algorithm of the tailored layer-optimized processing is an important part of the paper and aims to improve the detectability of IRHs. So, it is necessary to accurately assess the effectiveness of this algorithm in terms of the improvement in IRH detectability by comparing it with other methods that people have routinely been using in radar data processing. However, this paper does not have this kind of comparison and analysis, which is my major concern. The paper will be improved well for publication if this concern can be addressed.

> **Author response**:
> Dear referee #1,
>
> we thank you for your time to review our manuscript and your constructive feedback. Following your comments, we will provide a comparison of our implementation of LO-SAR processing to a moving averaging filter and to LO-SAR processing as implemented by Castelletti et al. (2019) as a supplement. In the manuscript we will highlight the differences and advantages compared to (Castelletti et al., 2019) more clearly. Finally, we include a short discussion about the selection of the slope iteration range based on the limit for the occurrence of destructive interference.
>
> Kind regards,
> Falk Oraschewski
> on behalf of all co-authors.

**Specific comments, questions, and suggestions:**

1. For the across-saddle and upstream profiles, it is desirable to see an echogram for each profile that is generated by applying a moving averaging filter of the same aperture length on each trace in Fig. 5 (c) and Fig. 5 (e). This averaging is like unfocused SAR processing without along-track decimation, which is much faster than the proposed LO-SAR processing. Because the phase shift due to slope has been corrected in reference phase substruction and thus the corrected phase is constant along the slope, this averaging is also similar to the coherent integration performed in the reference by Castelletti et al. (2019), i.e., the summation is not along the slope. By comparing these two echograms with Fig. 5 (d) and Fig. 5 (f),

the improvement on IRH detectability by the proposed LO-SAR can be qualitatively and quantitatively analyzed and discussed.

**Author response**: We agree that a direct comparison with existing processing methods will help to demonstrate the advantage of the proposed LO-SAR processing as implemented here for mobile pRES data. A figure with this comparison will be provided as a supplement to our manuscript.

In summary the advantages are as follows:

1. Moving averaging that includes the phase (and corresponds to unfocused SAR processing) will result in patterns of destructive interference even at moderate slopes (see following comment). Alternatively, non-coherent moving averaging could also be applied by only taking the power into account. But in this case the full potential of the phase-coherent signal cannot be used, which particularly lies in the desired extinction of spatially incoherent signals of non-specular reflections and of background noise.

2. Our implementation of LO-SAR processing does not substantially differ from the implementation by Castelletti et al. (2019), but provides some improvements in the processed radargram. To first highlight the differences, Castelletti et al. coherently sum along a range bin over some aperture length whereby they iterate over a range of phase shifts to find the optimal signal-to-noise ratio (SNR). Thereby, their approach provides the englacial slopes as a byproduct computed from the optimal phase shift. Their implementation to our data would correspond to applying LO-SAR processing on the raw signal before the reference phase correction by finding a phase-shift that cancels out the horizontal phase-gradient.

Our approach can be understood as an implementation of the LO-SAR processing by Castelletti et al. tailored to mobile pRES data. In this regard, we see a central contribution in demonstrating that the corrected phase is constant along IRHs which can be used to directly extract their slopes by iterating over a range of slopes. We describe this in the manuscript in detail not least because it helps to clarify common misunderstandings regarding the information content of the corrected phase signal and thereby has relevance for the interpretation of other types of pRES data. Only in a second step the obtained slopes are then used to perform the coherent summation, which has two advantages: (i) The slopes can be filtered prior to the coherent summation to remove outliers, for example occurring at the transition between two reflections. (ii) Coherent summation can be performed directly along the englacial slope, which for any point on the radargram (whether there is an IRH or not) reduces the integration of signal power from other nearby IRHs. In combination this results in some improvements, as for example a better resolution of closely spaced IRHs. We will revise the LO-SAR processing section of the manuscript to clearer highlight these differences.

2. In 5.2 for discussions on slope estimation, the authors mentioned that "strong reflections spread out over several range bins, across which the raw phase tends to be stable", and according to Fig. 5 (b), the "inclination relative to the surface is quantified during the LO-SAR processing and attains values of about 10° at both ends". For the aperture length of 5

meters used in the processing, this inclination corresponds to only two range bins from one end of the aperture to the other end, therefore there might be no significant difference in performing the summation along the slope and not along the slope if the reflected power does not have big difference neither within two range bins. It would be helpful to demonstrate at what aperture length and at what slope angle the improvement of SNR for IRH detectability can be expected from the proposed LO-SAR.

> **Author response**: The two cited statements are not in contradiction with the improvement by LO-SAR processing, because the first statement refers to the raw phase, whereas the summation is applied to the corrected phase. The raw phase is approximately stable in the vertical across a reflection, but varies horizontally (as the horizontal gradient of the raw phase is proportional to the slope of the reflection). The corrected phase, on the other hand, strongly varies in the vertical and in the horizontal is only constant along the slope of the reflection (Fig. 4b). Accordingly, horizontal summation that does not take (even small) slopes into account will result in destructive interference using either of them.
>
> The limit for the occurrence of destructive interference can be estimated from the phase offset between range bins of $\frac{3}{p}\pi$ (Eq. 4). With a zero-padding of $p = 8$ some points along the aperture length will be completely out of phase when the gradient of the summation line deviates from the correct reflection slope by 3 range bins (after zero-padding) along the aperture length. For an aperture length of 5 m this corresponds to a slope of $\sim 2°$ which is present at large parts of the radargram.
>
> In addition, note that the first statement is already true without zero-padding, but more pronounced in, and here used while referring to, the zero-padded signal for which with $p = 8$ a slope of $10°$ would correspond to 16 range bins. Nevertheless, zero-padding does not generally affect the destructive interference limit as the phase offset vs. total range is independent of it.

3. The iteration range of slopes used was from $-30°$ to $30°$ in steps of $0.2°$, much larger than the slop range observed in the data. Some discussion about smart selection of this range may be included in section 5.2 for reducing the computation intensity of the proposed LO-SAR.

> **Author response**: Thank you for this nice suggestion. We will include a short discussion about the selection of the iteration range for the slopes into the manuscript. This selection essentially depends on two factors: the step size should lie well below the limit for the occurrence of destructive interference (see above), and the maximum range should cover the range of slopes that are to be expected. We decided to use a rather wide slope range to not *a priori* prevent the detection of potential stronger inclined reflections in the near-basal ice. In our case compute time was not a limiting factor, because the profile length is short, but in other applications a better choice of the iteration limits will surely be useful.

4. What was the data logger sampling rate used during data collection? 40 kHz as discussed in section 5.1?

**Author response**: Yes, we used the sampling rate of the pRES data logger of 40 kHz. Technically, the ADC on the VAB board of the pRES samples at 80kHz and then two consecutive samples are averaged to give a 40kHz output which is written to the SD card. The pRES can be set to use the full 80kHz sampling rate, but because the AF filter is attenuating most of the additional high frequency input anyway, this would not be of much help.

**Technical corrections and suggestions:**

1. In Fig. 2 (a), the two legends are hard to distinguish, consider changing one of them with dashed line. The two red line segments for antennas need to explicitly be mentioned either in the figure caption or in text.

   **Author response**: We agree regarding the legends and will adapt the line style. The two red segments originate from another use of the graphic antenna models and will be removed.

2. In Fig. 3, missing information flow direction arrow along the line between the block for "Pre-processed mobile pRES data" and the block for "Moving median filtering of S".

   **Author response**: These arrows were supposed to indicate that the "Estimated slopes S" and the "Power and phase of traces in synthetic aperture length" are used together in the "Coherent summation of power along S". We will include additional arrow heads to make clear that information is not directly flowing between "Pre-processed mobile pRES data" and "Moving median filtering of S"

3. Revise "by the weighting the values" to "by weighting the values" at line 166 on page 8.

   **Author response**: Fixed.

4. It is suggested to mark the crossover between the cross-saddle profile and the GPR profile, the crossover between the cross-saddle profile and the upstream profile towards KCC, and the crossover between the upstream profile towards KCC and the GPR profile with A, B and C respectively in Fig. 1, and according to mark those vertical white lines in Fig. 5 with A, B and C. It is much easier this way to identify these crossovers.

   **Author response**: Thank you for this suggestion, we agree that this improves the clarity of the crossover points and will adopt it.

5. The location of the white line to the right in Fig. 5 (a) does not match with the location of the crossover between the upstream profile towards KCC and the GPR profile in Fig. 1 which is at the very end of the GPS profile.

**Author response**: Thank you for catching this mistake. Indeed, there was an error in the plotting range of Fig. 5, which we have fixed.
* * *
**References**

Castelletti, D., Schroeder, D. M., Mantelli, E., and Hilger, A.: Layer Optimized SAR Processing and Slope Estimation in Radar Sounder Data, Journal of Glaciology, 65, 983–988, https://doi.org/10.1017/jog.2019.72, 2019.

---

## Author Comment (AC2)

**Response to referee Benjamin Hills on egusphere-2023-2731**

**Summary.**

Oraschewski et al. describe a novel setup and implementation of the popular ApRES instrument, specifically for SAR profiling in alpine environments. Overall, I believe that this work could be highly impactful, particularly for alpine glaciology studies. As I describe in detail below, I believe that some additions to the engineering discussion as well as some organizational changes could elevate the impact of this article. It is well written and a great fit for *The Cryosphere* after these minor revisions.

> **Author response**:
> Dear Dr. Benjamin Hills,
>
> we thank you for your thorough review and constructive comments and suggestions on our manuscript. In particular, we appreciate and agree with your suggestion that segregating more clearly between the engineering and the science aspects of the manuscript will increase its impact. We will address this by merging the proof of concept into one section and revising the conclusion to focus more on the radar system design. We also agree that MIMO configurations, repeat measurements and polarimetry are interesting topics and relevant next steps in the context of our study. Following your suggestion, we decided to include them in a section about "Future directions". In addition, we link to a separate publication from our group (Ershadi et al., In press.) which addresses some of these topics in more depth.
>
> Kind regards,
> Falk Oraschewski
> on behalf of all co-authors.
>
> Note: In the following replies and our manuscript we refer to the autonomous phase-sensitive radio echo sounder (ApRES) only as pRES, because the presented survey type is far from being autonomous. A similar note will also be included in the manuscript.

**General Comments.**

I love this, neat design and I learned a lot. My largest high-level comment is that the article, as written, feels like it is in between a science narrative and an engineering narrative. I believe that you are intending for this to be read as an engineering article with a scientific proof of concept, and I think that some very minor organizational changes could help make that clearer. I would completely segregate the engineering language from the site-specific scientific proof of concept, preferably loading the engineering up front. I would also generalize the engineering discussion as I describe in paragraphs below. Ideally, anyone who is thinking about using a pRES to do profiling in future studies will be coming back to this article to figure out how to optimize their setup.

> **Author response**: You are right that this manuscript follows both an engineering approach of mobilizing the pRES and the scientific question of studying the deep stratigraphy of Colle Gnifetti. While the latter was the initial motivation for this study, the

manuscript now focuses mainly on the technical discussions which are more broadly relevant and applicable. We agree that a clearer segregation between the two aspects will be helpful for the readers. We will follow your suggestion and consolidate the proof of concept study in one section.

However, we believe that generalizing the engineering discussion is only partially helpful as the specifications of the pRES (except for the chirp time which we extensively discuss) are pretty much confined to what we are using (see below). A further generalization to all FMCW radars, is similarly not feasible as we can only speculate about the design decisions taken in future instrument development. Instead, we aim to motivate and guide the future development of improved radar systems by demonstrating the potential of using lightweight and ground-based FMCW radars for profiling and clearly communicating the limitations of the pRES, which are not least expressed by the specification numbers. Yet, we will also expand on guiding future mobile pRES deployments, for example by adding considerations about antenna orientations and alternative GNSS receivers.

One plausible organization scheme could be:
1. Introduction
2. Hardware design & Data Acquisition
   - Continuous acquisition vs stop-and-go
   - MIMO (might be too much for this article?)
3. Data Processing
   - Standard FMCW
   - LoSAR
   - Multipass (might be too much for this article?)
   - Polarimetry (might be too much for this article?)
4. Proof of Concept at Colle Gnifetti
   - Site Description
   - Radar Dataset
   - Comparison to ice core (I would put *all* the chemistry background, results, and discussion in this one section, it is interesting but not the focus of your manuscript and confusing when it comes up in three separate places)
5. Discussion
   - Mainly covers the nuance of your proof of concept (which is what you are already doing with the Discussion).
6. Conclusion

As you see in my outline above, I added a few sections which you do not currently cover (e.g. MIMO, multipass, and polarimetry, I expand on each in the following paragraphs). Here, I am encouraging you to consider broader use cases for this instrument rather than limiting the conversation to what you end up doing in the example, that way you can have a bit broader impact with this article. You could also consider adding a section (possible between Discussion and Conclusion?) on "Future Directions" where you would put some of this multipass and polarimetry language.

Sections 2 and 3 obviously have a lot of overlap and may need some consideration in how they are folded together or not.

**Author response**: Thank you for your helpful suggestions on the general structure of the manuscript. We will in particular adopt the suggestion of merging the proof of concept study into one section and include a section on "Future directions" (see also the detailed replies below). Moreover, we aim to improve the guidance of the reader by separating between hardware design and data acquisition and by including some additional links between the sections.

*ApRES specificity*

I believe you are currently too specific to particular ApRES settings. In my opinion, the engineering language would be more useful if you generalized the numbers for center frequency, bandwidth, and chirp time (all of which can be changed in software as far as I know). Then, in your proof of concept section you give the specifics for that particular case and use those specifics as justification for your survey design (e.g. the stop and go surveying mode).

**Author response**: It is correct that the FMCW ramp of the pRES can be changed in the settings with the configurable parameters of the pRES being the start and end frequency of the FMCW ramp, the frequency step size (as the signal has essentially discrete steps) and the time that each step lasts. Besides, the frequency range can be cut in the post-processing by cutting the mixed signal before applying the spectral analysis (Vaňková et al., 2020). But except for the total chirp time which we discuss in the manuscript and will expand with a comment about increasing noise, the default settings of the pRES mostly exhaust the capabilities of the hardware. The upper and lower frequency limits set by the hardware are at about 100 and 450 MHz, but because the recorded power already decreases towards both ends, extending the frequency range towards these limits would not directly result in a corresponding improvement of the range resolution. In addition, the operation range of the skeleton slot antennas is also approximately confined to the 200-400 MHz range.

As we have no data that was recorded with an increased frequency range, but also do not expect a significant improvement in the range resolution, we want to avoid the impression that this can be expected. Therefore, we decided against generalizing the numbers of the bandwidth and center frequency as it is done in many other studies that use the pRES (e.g., Brennan et al., 2014; Vaňková et al., 2020; Kapai et al., 2022).

*MIMO and Polarimetry*

I know that this team has already been thinking about multi-input-multi-output designs, so I am eager to know what they think that might look like for this alpine setting? Is that infeasible here? Too heavy to carry by ski? My tendency would be to include this, even if it is presented as a possible next direction, then go more specific to your single-channel design in the proof-of-concept section.

If you include MIMO in the hardware section you would want to at least mention how the multiple channels helps you, whether that be variable offset (for velocity inversions) or mult-polarization for polarimetry.

**Author response**: Our team has indeed been working on MIMO and polarimetry designs, which are addressed in a separate publication about rover-towed quad-polarimetric radar measurements which has just been accepted for publication (Ershadi

et al., In press.). We will refer to it for a detailed discussion of mobile polarimetry measurements.

For the main application addressed in this manuscript of profiling of mountain glaciers, we do not think that MIMO approaches with the pRES are feasible, because the system would become to heavy to be towable by hand. Already with two antennas a second person was mostly needed to help in stabilizing the system or even pushing it at the upstream profile. We think that further instrument development for FMCW systems that operate in a higher frequency range and, thus, with smaller and lighter antennas is necessary to make such survey types feasible. We will therefore not include a description of how to process these type of data in the methods section, but include a discussion of potential advantages to argue for such development efforts in the future.

*Multipass*
This is the other obvious use case for your design which would not need any hardware changes. If you left out MIIMO and polarimetry that seems fine to me, but I think this should be included because in my mind it is an obvious use case for your instrument. Specifically, what I mean is resurveying the same profile between two different times to measure vertical velocities.

> **Author response**: We like the idea of using the mobile pRES for repeat surveys and will include a discussion of this potential application in the manuscript. But as we do not have such data from different times, we are not able to fully assess and demonstrate what efforts are needed to sufficiently correlate such repeat surveys. Our data could give a first impression for such efforts, because we have performed a two-fold direct repeat survey along an approximately 30 m section of the across saddle transect. However, this data does not help in assessing how to deal with changing englacial reflection characteristics, changing vertical positioning, horizontal positioning uncertainty and potential changes in the radar setup. Moreover, we regard this section as too short for clearly demonstrating the general feasibility. Therefore, we will leave a detailed development of processing routines for repeat surveys to future studies.

**Specific Comments.**

Title – I would say it is a proof-of-concept at Colle Gnifetti. The current title makes it sound like this instrument is only useful at one site.

> **Author response**: Agreed, thank you for the suggestion. We will adjust the title to "Layer-optimized SAR processing with a mobile phase-sensitive radar: a proof of concept for detecting the deep englacial stratigraphy of Colle Gnifetti, Switzerland/Italy".

Abstract – I believe that "(pRES)" and "(LO-SAR)" don't need the acronym in parentheses since you only use them once in the abstract and you redefine in the text.

> **Author response**: We have removed "(LO-SAR)", but keep "(pRES)" as we are now using the acronym in the implemented changes for the comment on L10.

L10 – "down to the base..." you could be more specific. "Improved from 50% depth with pulsed radar to 80% depth with lo-sar"?
*Coming back to this since in the discussion you mention the ages that you resolve (78 yr vs 288 yr) I think those are useful numbers to have in the abstrace if it is something you want to emphasize.*

> **Author response**: Thank you for the suggestion. We have included these number into the abstract, where we now write: "Compared to previously deployed pulsed radar systems, with the mobile pRES the age of the oldest continuously traceable IRH could be increased from $78 \pm 12$ a to $288 \pm 35$ a.

"
L25 – Instead of the in press article, there are many alternatives you could cite (Arcone et al., 2005; Medley et al., 2013).

> **Author response**: The in press article is published by now so that we decided to keep it. In addition, we will include a reference to the equally relevant and more recent work of Cavitte et al. (2018).

L34 – "neither of which are applicable..."

> **Author response**: Fixed.

L57 – Instead of "invisible" I might say "unresolved"

> **Author response**: Fixed.

L85 – I recently talked with Keith Nicholls about doing pRES profiling and he suggested that I shorten the chirp time. I didn't have a lot of luck with it, but my experience has me thinking about whether you want to change how this gets presented here. If you give some guidelines for how short a chirp you need to profile continuously the community may come back to this paper a lot more.
*I am now seeing that you do talk through some of this in the discussion, but I think talking about it when it is introduced will be helpful (I am guessing that a lot of readers will be primarily interested in your hardware design and processing flow).*

> **Author response**: After careful consideration of moving these technical discussions into the methods section, we decided to keep our current approach of focusing in the methods section on what we have actually done. These discussions do not only build upon later chapters, but also are more extensive and might easily distract from the applied methodology. Instead, we decided to include a statement at the end of the methods section that redirects the interested reader directly to these discussions.

LL102-103 – as I said before, it might be more broadly useful to the community to generalize some of the numbers here until you get to your proof-of-concept where you would give specifics.

> **Author response**: See comment on "ApRES specificity".

LL113-115 – I might add 'volumetric scattering' in line with my comment above on adding another adjustment.

> **Author response**: Unfortunately it is at the moment unclear to us to which sentence this comment is referring to.

LL134-139 – This is fine to do here in my opinion, but I think it is not precisely correct since you are compressing information over the full bandwidth to get the range in eq. 1. I might just add a bit more nuance so people don't misuse this approximation.

> **Author response**: The deramped frequencies obtained from an FMCW radar signal by compressing information over its bandwidth essentially translate into two-way travel times and not ranges. Eq. 1, as introduced by Brennan et al. (2014) and commonly used, takes the short-cut of directly converting the deramped frequencies into ranges by assuming a constant permittivity (of ice). Here, we simply use a variable permittivity, analogous to the density correction of pulsed GPR data. Both approaches are justified because the permittivity of ice and air, and accordingly also firn, do not change considerably over the bandwidth of the pRES (Bohleber et al., 2012).

LL143-144 – I may be confused the "explicit sum", but I believe Castelletti et al. (2019) did something similar if not the same. They do a coherent sum in their eq. 3 and then "incoherent averaging" as described in the subsequent paragraph. Maybe you are doing something different; if so, more explanation would be helpful.

> **Author response**: We will make sure to clearer communicate the similarities and differences to Castelletti et al. (2019). A detailed explanation of these differences is given in response to referee #1. In summary, we are first estimating the slopes of IRHs using the corrected phase, which allows us to (i) filter the slopes before the coherent summation and (ii) sum directly along the IRHs. Castelletti et al. essentially use the raw signal and perform the coherent summation along range bins, where they optimize for a phase shift that counterbalances the horizontal phase gradient of the raw phase. Their approach provides the englacial slopes as a byproduct. The two implementations are similar, with ours resolving closely spaced reflections slightly better. However, in addition to this improvement, a central reason for introducing our implementation in detail is also that it offers insights into the correct interpretation of the corrected phase signal by demonstrating that it is constant along IRHs.

LL153-154 – This is interesting. I have actually had a lot of luck with the method from MacGregor et al. (2015) (it is much faster as you say), but that has been with MCoRDS data and I hadn't considered the phase uncertainty in this way. I am interested in seeing a comparison figure although I get that most readers are not me and this is not the emphasis of the article, so disregard unless you think it is useful.

> **Author response**: We agree that the approach by MacGregor et al. (2015) is generally very useful, which is why we state that the limitations only apply "in our data". The

phase uncertainty in our survey is probably significantly higher than in most MCoRDS data. This can among other things be attributed to the fact that no stacking could be applied due to the high acquisition time, to the more complicated geometry of a mountain glacier which causes higher spatial signal variations and to the vicinity to sources of interfering signals. Accordingly, to filter out the resulting uncertainties some spatial averaging is needed for our data which is provided by the LO-SAR approach.

LL159-176 – I believe this is mostly the same as Castelletti et al. (2019), correct? Either way, I think it is important for you to write it out here (especially since the way you write it is a bit different). Still might be worth citing them and pointing out explicitly where you are (or are not) different.

**Author response**: See comment to LL143-144, we will clearer communicate the similarities and differences to Castelletti et al. (2019) and cite them again when discussing the details of our implementation.

LL181 – comma feels like it is in a weird spot to me

**Author response**: Fixed.

L212 – I believe more details on "poorer data quality" could be exceedingly useful for future studies trying to do this type of thing.

**Author response**: We gave these details in LL321-327. With the merging of the proof of concept section, we will bring the two statements together.

L214-215 – save mention of the ice core for the next section

**Author response**: This will be adopted with the reorganization of the manuscript.

L230-231 – I believe this result is correct. You might add a citation to Greenland radar work and the Holocene-glacial contrast, longer time period but similar idea.

**Author response**: You are right, we will include a reference to (Karlsson et al., 2013) to support our interpretation.

L248-249 – Nice. "should not be a more painstaking data collection using pulsed GPRs but rather aim at accelerating the profiling capabilities of FMCW radars". Very much agree with this.

**Author response**: Thank you.

L 274-278 – What is required to change the frequency band. I think you can change it in software, but I am guessing the skeleton slots are specifically designed for this band. Could someone design new antennas and use the same unit? As Jonathan has been doing for the HF ApRES but now to a higher frequency band? Could be some interesting additional notes here.

**Author response**: See comment on "ApRES specificity". Extending the frequency band as well as shifting it requires a complete redesign of the hardware. Besides the skeleton slot antennas which are approximately limited to the 200-400 MHz range, other components of the radar system such as the RF amplifiers/filters and the AF filter also need to be adjusted.

L307 – most likely "not"

**Author response**: Fixed.

L328-335 – Do you think there is also a lot of uncertainty in the density which causes thickness uncertainty? Or you are decently confident on the density?

**Author response**: We are decently confident that the uncertainty in the density cannot explain a 8 m deviation in the range between ice core length and bed reflection depth in the radar data. We estimated that the density uncertainty causes a range uncertainty on the order of 1 m (see Section 3.3).

This was done by applying the density correction using the independently and differently measured KCC and CG03 firn density data and extracting the predicted range difference between the two. This approach is justified, because uncertainties in the density at Colle Gnifetti mainly originate from its spatial variability due to the strong local variability in snow deposition (Alean et al., 1983), where CG03 represents the firn conditions on the saddle and KCC represents the conditions in the Southern flank, which well cover our study area. The deviation between the two datasets is higher then the standard deviation of the single density profiles (and in particular of the KCC data which was measured with high vertical resolution). We therefore consider our density estimate to be rather conservative.

L336 – I would add that it improves SNR specifically for specular reflectors, which the bed is not in most cases.

**Author response**: You are right, stating that LO-SAR processing is "improving the SNR of the radargram", was imprecise and we change the sentence to "improving the SNR of specular reflections in the radargram".

L349 – "promote" changes in the COF? Or "caused by" changes in the COF?

**Author response**: Here, we try to distinguish between the cause (impurities) and the mechanism (acidity/COF changes) for deep IRHs. It was meant that impurities might promote (localized) changes in the COF. We now write "Accordingly, deep IRHs are caused by impurities, which determine the acidity and perhaps may induce localized gradients in the COF.", which we hope is more clear.

L368 – This group could probably convince me (so please do if I am wrong!) but I don't think it is the impurity layers which are causing changes in the COF. Rather that boundaries between two different COFs can cause a reflection.

**Author response**: We agree that boundaries between two different COFs/sharp gradients in the COF can cause reflections, which is not contradicting with our statement. Here we say that horizons of such localized gradients in the COF might be induced by impurities in the first place. This is for example hypothesized in (Drews et al., 2012) as a cause for higher anisotropic backscatter in glacial ice and might be explained by an impurity control of the grain size (Kerch, 2016).

Conclusion – Too focused on your site. I believe that this radar design will be highly useful to many groups all over the world, do not undersell yourself by making the reader thing it is only for this site!

**Author response**: Agreed. We will revise the conclusion to reduce the Colle Gnifetti specific content and focus more on the radar system and insights from the signal processing.

L388 – Excited to see the processing scripts when you have them up! Please do be sure to post them.

**Author response**: The scripts are now available at https://github.com/oraschewski/pRocESsor. For the final published version of the manuscript a permanent release of this repository will be created.

**Figures**

Figure 1. I am not very familiar with this area so an inset map showing where you are located in the alps and then where on the glacier, those would be helpful. As with my general comments, I think you should move this figure lower. Having Figure 1 be a map makes it feel like this is a scientific paper, whereas if you want it to feel more like an engineering paper starting with Figures 2 and 3 would be helpful.

**Author response**: As suggested, we move the current Figures 2 and 3 in front of Figure 1 and include an inset map.

Figure 3. If you decide to include language on MIMO and multipass then I would add more to this flowchart as well.
I was at first confused about the arrow labeled "power and phase of traces in synthetic aperture length". I didn't understand the direction. I would put little arrowheads along that line to help a reader out.

**Author response**: We add additional arrowheads to the paths which point together towards "Coherent summation of power along S". However, we decided against including any additional content in this flowchart to avoid additional confusion and, in particular, because we do not have MIMO or repeat data from a later time, we can only speculate about what additional steps will be needed to successfully correlate and process these data and how this is best done.

Figure 4. It is tough to see the layers with your red annotation lines over top of them. Either drop the lines and label with points/arrows (as you do in e and f) or make additional panels for interpretation as in Welch and Jacobel (2005) Figures 3a and 5b.

> **Author response**: We agree with your concern that we are masking these layers. We prefer to keep the lines in this figure as the differences are otherwise still difficult to see in the figure size that fits into the manuscript. Instead, we will include a zoom into these sections in the supplementary material.
* * *
**References**

Alean, J., Haeberli, W., and Schädler, B.: Snow Accumulation, Firn Temperature and Solar Radiation in the Area of the Colle Gnifetti Core Drilling Site (Monte Rosa, Swiss Alps): Distribution Patterns and Interrelationships, Zeitschrift für Gletscherkunde und Glazialgeologie, 19, 131–147, 1983.

Arcone, S. A., Spikes, V. B., and Hamilton, G. S.: Stratigraphic Variation within Polar Firn Caused by Differential Accumulation and Ice Flow: Interpretation of a 400 MHz Short-Pulse Radar Profile from West Antarctica, Journal of Glaciology, 51, 407–422, https://doi.org/10.3189/172756505781829151, 2005.

Bohleber, P., Wagner, N., and Eisen, O.: Permittivity of Ice at Radio Frequencies: Part II. Artificial and Natural Polycrystalline Ice, Cold Regions Science and Technology, 83–84, 13–19, https://doi.org/10.1016/j.coldregions.2012.05.010, 2012.

Brennan, P. V., Lok, L. B., Nicholls, K., and Corr, H.: Phase-sensitive FMCW Radar System for High-precision Antarctic Ice Shelf Profile Monitoring, IET Radar, Sonar & Navigation, 8, 776–786, https://doi.org/10.1049/iet-rsn.2013.0053, 2014.

Castelletti, D., Schroeder, D. M., Mantelli, E., and Hilger, A.: Layer Optimized SAR Processing and Slope Estimation in Radar Sounder Data, Journal of Glaciology, 65, 983–988, https://doi.org/10.1017/jog.2019.72, 2019.

Cavitte, M. G. P., Parrenin, F., Ritz, C., Young, D. A., Van Liefferinge, B., Blankenship, D. D., Frezzotti, M., and Roberts, J. L.: Accumulation Patterns around Dome C, East Antarctica, in the Last 73 Kyr, The Cryosphere, 12, 1401–1414, https://doi.org/10.5194/tc-12-1401-2018, 2018.

Drews, R., Eisen, O., Steinhage, D., Weikusat, I., Kipfstuhl, S., and Wilhelms, F.: Potential Mechanisms for Anisotropy in Ice-Penetrating Radar Data, Journal of Glaciology, 58, 613–624, https://doi.org/10.3189/2012JoG11J114, 2012.

Ershadi, M. R., Drews, R., Hawkins, J., Elliott, J., Lines, A. P., Koch, I., and Eisen, O.: Autonomous Rover Enables Radar Profiling of Ice-Crystal Fabric in Antarctica, IEEE Transactions on Geoscience and Remote Sensing, In press.

Kapai, S., Schroeder, D., Broome, A., Young, T. J., and Stewart, C.: SAR Focusing of Mobile ApRES Surveys, in: IGARSS 2022 - 2022 IEEE International Geoscience and Remote Sensing Symposium, pp. 1688–1691, IEEE, Kuala Lumpur, Malaysia, ISBN 978-1-66542-792-0, https://doi.org/10.1109/IGARSS46834.2022.9883784, 2022.

Karlsson, N. B., Dahl-Jensen, D., Gogineni, S. P., and Paden, J. D.: Tracing the Depth of the Holocene Ice in North Greenland from Radio-Echo Sounding Data, Annals of Glaciology, 54, 44–50, https://doi.org/10.3189/2013AoG64A057, 2013.

Kerch, J. K.: Crystal-Orientation Fabric Variations on the Cm-Scale in Cold Alpine Ice: Interaction with Paleo-Climate Proxies under Deformation and Implications for the Interpretation of Seismic Velocities, https://archiv.ub.uni-heidelberg.de/volltextserver/22326/, https://doi.org/10.11588/heidok.00022326, 2016.

MacGregor, J. A., Fahnestock, M. A., Catania, G. A., Paden, J. D., Prasad Gogineni, S., Young, S. K., Rybarski, S. C., Mabrey, A. N., Wagman, B. M., and Morlighem, M.: Radiostratigraphy and Age Structure of the Greenland Ice Sheet, Journal of Geophysical Research: Earth Surface, 120, 212–241, https://doi.org/10.1002/2014JF003215, 2015.

Medley, B., Joughin, I., Das, S. B., Steig, E. J., Conway, H., Gogineni, S., Criscitiello, A. S., McConnell, J. R., Smith, B. E., van den Broeke, M. R., Lenaerts, J. T. M., Bromwich, D. H., and Nicolas, J. P.: Airborne-Radar and Ice-Core Observations of Annual Snow Accumulation over Thwaites Glacier, West Antarctica Confirm the Spatiotemporal Variability of Global and Regional Atmospheric Models, Geophysical Research Letters, 40, 3649–3654, https://doi.org/10.1002/grl.50706, 2013.

Vaňková, I., Nicholls, K. W., Xie, S., Parizek, B. R., Voytenko, D., and Holland, D. M.: Depth-Dependent Artifacts Resulting from ApRES Signal Clipping, Annals of Glaciology, 61, 108–113, https://doi.org/10.1017/aog.2020.56, 2020.

Welch, B. C. and Jacobel, R. W.: Bedrock Topography and Wind Erosion Sites in East Antarctica: Observations from the 2002 US-ITASE Traverse, Annals of Glaciology, 41, 92–96, https://doi.org/10.3189/172756405781813258, 2005.

---

## Author Response (AR2)

**Response to Editor Joe MacGregor's second review on egusphere-2023-2731**

Dear Editor Dr. MacGregor,

thank you for thoroughly reviewing our MS again and for providing helpful suggestions for last improvements. We agree with all of your suggestions and hope that you find our adaptations satisfying. Thank you again for taking your time to edit this MS, we believe that this process significantly improved its quality.

Kind regards,
Falk Oraschewski
on behalf of all co-authors.
* * *
**Remarks**

My apologies for the delayed decision, but I have now had a chance to review your revised MS. I consider it well organized, improved upon the original, and responsive to the reviewers' concerns. While the physical scale of the survey is small, the breadth of the technical accomplishment, the exposition on the technical methods, and the improved interpretation and connection with previous glaciological studies in this region are all excellent. I recommend to publish but have noted the following minor issues for you to address beforehand.

> **Author response**: Thank you for highlighting these strengths of our MS. We agree that the spatial coverage of this proof of concept is limited. This can not least be attributed to the technical difficulties and the current limitations of the mobile pRES system that we describe in the MS. Nevertheless, we believe that our results are promising and hope that this work will help to overcome the limitations to achieve a more extensive mapping of the deep stratigraphy of mountain glaciers in future surveys.

The link to the raw data (https://doi.org/PANGAEA.965199) does not work.

> **Author response**: You are right, I made a mistake when copying the link. Now the correct link is provided.

35: "aircraft" not "aircrafts"

> **Author response**: Fixed.

37-38: The following phrasing makes more sense: "Here we address the need for a lightweight...for SAR processing."

> **Author response**: Agreed. Thank you for this suggestion.

65 and elsewhere in the MS: When referring to specific values, unit abbreviations are fine (e.g., "5.5 cm"), but when referring to a unit for a sense of scale, they should be spelled out, i.e., decimetre for dm, centimetre for cm.

**Author response**: Fixed

72: Clearer: "...antennas extends further along-track than cross-track and ground targets..."

**Author response**: Fixed. Thank you for the suggestion.

74-5: This final sentence is better suited to the discussion on potential software/hardware improvements.

**Author response**: Agreed. Because the previous sentences already introduce this discussion point, we decided to move them, as well. As this would make Section 5.1 quite long, we moreover decided to split it into two parts ("Feasibility assessment" and "System improvements").

76: Define GNSS acronym at first use.

**Author response**: Fixed.

80: I see what you mean but the phrasing that starts this sentence is awkward ("build upon" and "low-cost" are sort of opposing), and as for 74-5 this sentence on potential improvements likely belongs later on to better separate what was done from what could be done.

**Author response**: Thank you for pointing out this subtle meaning of "build upon". We have moved this sentence to the discussion and now write "In addition, to follow the low-cost approach of the pRES, we suggest to combine it with low-cost GNSS receivers that can achieve a positioning accuracy that is comparable to commercial instrumentation (Still et al., 2023; Pickell and Hawley, 2024)".

88: What type of signal was used to indicate to the operator the chirp was complete? Audible or visible?

**Author response**: The signals were audible. Different types of beep tones were used for indicating to start and stop moving. In addition, constant beeping indicated that the operator has moved too far.

90-2: Again, save this last statement for the discussion.

**Author response**: Agreed. We have removed the sentence and instead just point in the initial paragraph of Section 2 to the discussion of potential system improvements.

212: "systems" not "system"

Author response: Fixed.

225: "...data reveals deep IRHs..."

Author response: Fixed.

257: remove comma

Author response: Fixed.

257-261: Could the thickness decrease also be attributable to local thinning associated with recent climate warming?

Author response: No, we regard this option as highly unlikely. The glacier saddle and the upper parts of Grenzgletscher are still frozen at the base and do not show dynamic changes that could explain such strong thinning. Neither has snow precipitation reduced or surface melting increased enough to thin the glacier from the top. Moreover, we do not see any substantial differences in the two-way travel times of the bed reflections between the radar data recorded in 2008 and 2021.

260: "non-straight", i.e., "non-vertical"?

Author response: Fixed.

350: "less thick", i.e., "thinner"?

Author response: Fixed.

354-5: Perhaps clearer: "....  , a redesigned FMCW system is desirable that is similar to many airborne systems whose chirps are much shorter ($\leq$X us)."

Author response: Thank you for the suggestion, we have adapted it.

Figure 1: Could the E-plane direction mentioned the text be added here?

Author response: Yes, the E-plane direction is now included in the schematic figure.

Figure 3 caption: "example" not "exemplary"

Author response: Fixed.

Figure 5: This figure is great, but I think the range of panel b could be reduced to $\pm10°$without substantial information loss, and inequalities used to illustrate that sometimes the apparent slope is higher.

Author response: Agreed, we have adapted the color range of the figure now.

**References**

Pickell, D. J. and Hawley, R. L.: Performance Characterization of a New, Low-Cost Multi-GNSS Instrument for the Cryosphere, Journal of Glaciology, pp. 1–7, https://doi.org/10.1017/jog.2023.97, 2024.

Still, H., Odolinski, R., Bowman, M. H., Hulbe, C., and Prior, D. J.: Observing Glacier Dynamics with Low-Cost, Multi-GNSS Positioning in Victoria Land, Antarctica, Journal of Glaciology, pp. 1–18, https://doi.org/10.1017/jog.2023.101, 2023.